# 💿 $\mathcal{DVD}$-QUANT: DATA-FREE VIDEO DIFFUSION TRANSFORMERS QUANTIZATION

**Zhiteng Li**[1*], **Hanxuan Li**[2*†], **Junyi Wu**[1], **Kai Liu**[1], **Haotong Qin**[3], **Linghe Kong**[1],
**Guihai Chen**[1], **Yulun Zhang**[1], **Xiaokang Yang**[1‡]
[1]Shanghai Jiao Tong University, [2]Zhejiang University, [3]ETH Zürich

## ABSTRACT

Diffusion Transformers (DiTs) have emerged as the state-of-the-art architecture for video generation, yet their computational and memory demands hinder practical deployment. While post-training quantization (PTQ) presents a promising approach to accelerate Video DiT models, existing methods suffer from two critical limitations: (1) dependence on computation-heavy and inflexible calibration procedures, and (2) considerable performance deterioration after quantization. To address these challenges, we propose $\mathcal{DVD}$-Quant, a novel Data-free quantization framework for Video DiTs. Our approach integrates three key innovations: (1) **Bounded-init Grid Refinement (BGR)** and (2) **Auto-scaling Rotated Quantization (ARQ)** for calibration data-free quantization error reduction, as well as (3) $\delta$-**Guided Bit Switching ($\delta$-GBS)** for adaptive bit-width allocation. Extensive experiments across multiple video generation benchmarks demonstrate that $\mathcal{DVD}$-Quant achieves an approximately $2\times$ speedup over full-precision baselines on advanced DiT models while maintaining visual fidelity. Notably, $\mathcal{DVD}$-Quant is the first to enable W4A4 PTQ for Video DiTs without compromising video quality. Code and models will be released to facilitate future research.

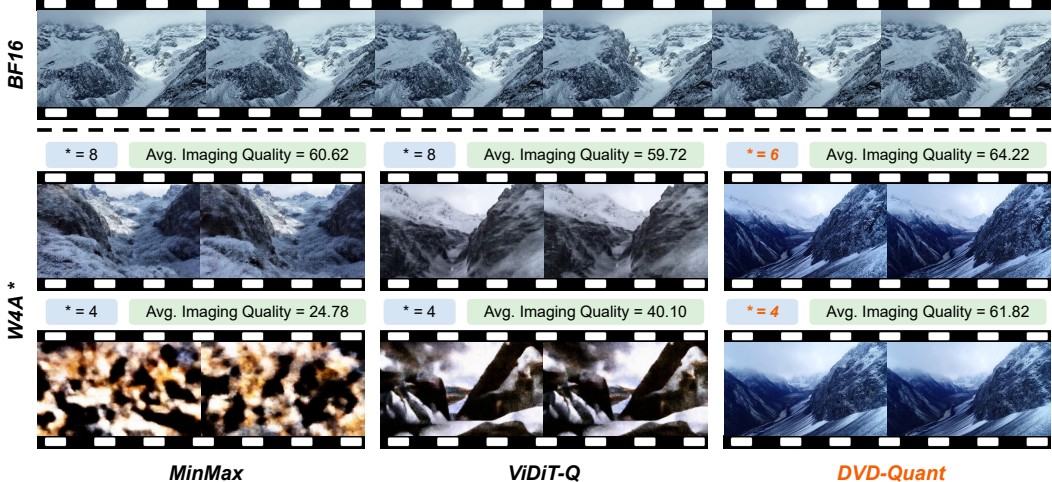

Figure 1: $\mathcal{DVD}$-Quant generates high-fidelity videos under both W4A6 (mixed-precision) and W4A4 settings, while baseline methods fail under low-bit activation quantization. $\mathcal{DVD}$-Quant remains effective even in such extreme scenarios.

## 1 INTRODUCTION

Recent advances in diffusion transformers (DiTs) (Peebles & Xie, 2023) have revolutionized video generation (Singer et al., 2023), enabling high-fidelity synthesis through iterative denoising processes. Innovations such as Sora's unified framework (OpenAI, 2024) and SkyReels-V2's infinite-length generation (Chen et al., 2025a) demonstrate enhanced controllability, while efficiency gains emerge from quantization (Zhao et al., 2025), sparse attention mechanisms (Xi et al., 2025; Zhang et al.,

---

[*]Equal contribution
[†]Work done during an internship at Shanghai Jiao Tong University
[‡]Corresponding author: Xiaokang Yang, xkyang@sjtu.edu.cn

2025) and cache techniques (Wu et al., 2025; Liu et al., 2024a). The emergence of large-scale models like HunyuanVideo (Kong et al., 2024) has further improved video generation quality with its causal 3D VAE architecture and full-attention mechanisms, achieving film-grade quality and physics-aware generation. These developments collectively demonstrate enhanced controllability in temporal coherence and resolution, though challenges persist in computational efficiency.

Prior work in diffusion model quantization has approached these challenges through two distinct paradigms. **Quantization-Aware Training (QAT)** requires full model fine-tuning but achieves superior low-bit performance. Ter-DiT (Lu et al., 2024) introduces RMSNorm-enhanced adaLN modules for stable ternary training of Diffusion Transformers. In contrast, **Post-Training Quantization (PTQ)** methods (Li* et al., 2025; Chen et al., 2025b; Zhao et al., 2025) offer deployment-friendly solutions without retraining. SVDQuant (Li* et al., 2025) employs low-rank SVD decomposition to absorb outliers into 16-bit branches for 4-bit weight/activation quantization on FLUX.1 models (Labs, 2024). ViDiT-Q (Zhao et al., 2025) introduces a set of techniques for DiT-based generative models and achieves negligible performance drop under W8A8 quantization. These approaches demonstrate the trade-off between QAT's higher accuracy and PTQ's faster deployment in DiT compression.

Although PTQ offers a plug-and-play alternative, existing PTQ methods still face two critical limitations. **First**, the calibration-based pre-scaling adopted by most video-specific quantization techniques (Zhao et al., 2025; Wu et al., 2024; Chen et al., 2025b) not only demands extensive calibration time but also struggles to adapt to timestep-dependent scale variations in DiTs. **Second**, aggressive W4A4 quantization results in significant performance degradation (Fig. 1), with VBench metrics dropping by **27.5%** to **61.3%**. Our analysis reveals three key insights to overcome these limitations: **(i)** Weights exhibit Gaussian-like distributions (Fig. 3), making fixed quantization ranges suboptimal for preserving critical parameters. **(ii)** Activation scales vary significantly across denoising timesteps, necessitating dynamic rather than static quantization strategies. **(iii)** Latent feature variations exist across diffennent denoising timesteps, enabling the possibility of adaptive bit-width allocation during online inference.

Building on these insights, we propose $\mathcal{DVD}$-Quant, a comprehensive quantization framework tailored for DiTs. To approximate the Gaussian-like weight distribution, we propose an iterative grid refinement strategy, which progressively adjusts the quantization scale and zero-point starting from their bounded-search initialization. For handling the timestep-variant activations scales, we combined online scaling with Hadamard rotation. Compared to calibration-based pre-scaling, it can achieve notably lower quantization error. Furthermore, to adapt to the latent feature variations across denoising steps, we propose a mechanism that automatically allocate appropriate bit-widths for activations at different timesteps. Our contributions can be summarized as follows:

- We conduct a systematic analysis of quantization challenges in large-scale Video DiTs, identifying three key characteristics: Gaussian-like weight distributions, substantial activation scale discrepancies, and latent feature variations across denoising timesteps.

- We propose **Bounded-init Grid Refinement (BGR)**, a novel weight quantization scheme for Gaussian-like distributions. By iteratively refining quantization grid upon tightening bounds, BGR significantly reduces the quantization error compared to fixed-range methods.

- We propose **Auto-scaling Rotated Quantization (ARQ)**, a calibration-free activation quantization method designed to address timestep-dependent scale variations. Leveraging online scaling with Hadamard rotation, ARQ maintains high model accuracy without requiring extensive calibration procedures.

- We propose $\delta$-**Guided Bit Switching**, an adaptive temporal-wise mixed-precision mechanism that allocates activation bit-widths by leveraging timestep feature variations. This mechanism optimizes bit allocation while incurring negligible additional inference overhead.

## 2 RELATED WORKS

### 2.1 DIFFUSION TRANSFORMERS (DIT)

Diffusion Transformers (DiTs) (Peebles & Xie, 2023) represent a paradigm shift in generative modeling, replacing the traditional U-Net (Chitwan Saharia, 2022; Andreas Blattmann, 2023; Aditya Ramesh & Chen, 2022) backbone with transformer architectures. This breakthrough was attributed to the self-attention mechanism's ability to model long-range dependencies (Vaswani et al., 2023), proving especially beneficial for high-resolution synthesis.

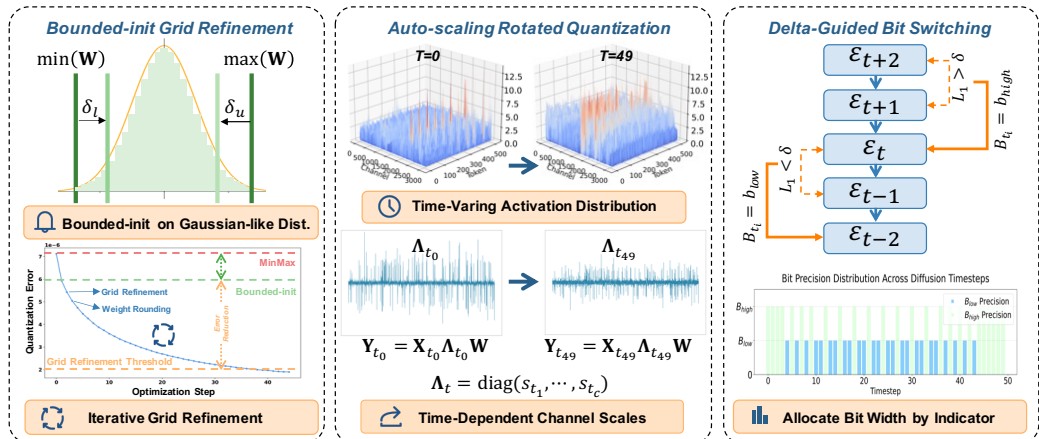

Figure 2: Overview of $\mathcal{DVD}$-Quant. Bounded-init Grid Refinement and Auto-scaling Rotated Quantization are data-free methods designed to reduce quantization errors for weights and activations, respectively. $\delta$-Guided Bit Switching adaptively assigns bit-widths to different time steps.

The success of DiTs in image generation quickly extended to video synthesis. Latte (Ma et al., 2025) pioneered this transition by adapting the DiT framework for temporal modeling, achieving unprecedented coherence in text-to-video generation. This was followed by Sora (OpenAI, 2024), which scaled DiTs to massive parameter counts and dataset sizes, demonstrating remarkable capabilities in long video generation with complex dynamics. Open-source implementations like Hunyuan-Video (Kong et al., 2024), Open-Sora (Zheng et al., 2024; Peng et al., 2025) and CogvideoX (Yang et al., 2024) further democratized these advancements, making them widely accessible.

Despite their impressive results, DiTs inherit fundamental challenges from both transformer architectures and diffusion processes. Attention's quadratic complexity is particularly problematic for video generation, where sequence length scales with spatial resolution and frame count. Moreover, the iterative nature of diffusion models requires multiple forward passes through the entire architecture (typically 50-100 steps), with each step processing increasingly refined features. These limitations have driven efforts to boost DiTs' efficiency, such as caching (Wu et al., 2025; Liu et al., 2024a), and quantization (Li* et al., 2025; Chen et al., 2025b; Zhao et al., 2025).

## 2.2 MODEL QUANTIZATION

Model quantization (Xiao et al., 2023; Li* et al., 2025; Liu et al., 2024b; Li et al., 2023a) has emerged as a fundamental technique for compressing deep neural networks (Li et al., 2025b;e;d;a; 2023d; 2025c; 2023c;b) by converting full-precision parameters into low-bit representations, significantly reducing memory footprint and computational costs. Post-Training Quantization (PTQ) (Li et al., 2025f; Frantar et al., 2023; Yan et al., 2025) has proven particularly effective as it compresses pre-trained models without requiring extensive retraining.

Significant progress has been made to address quantization challenges in transformers (Vaswani et al., 2017). SmoothQuant (Xiao et al., 2023) introduces channel-wise scaling to balance weight and activation quantization difficulty, while Quarot (Ashkboos et al., 2024) employs orthogonal matrix rotations to distribute values evenly across channels. For diffusion models, quantization presents unique challenges due to their time-dependent nature. Q-Diffusion (Li et al., 2023a) and PTQ4DM (Shang et al., 2023) address this by collecting timestep-wise activation statistics to determine optimal quantization parameters. Q-DiT (Chen et al., 2025b) tackles channel-wise imbalance via customized quantization parameters. Subsequent work such as SVDQuant (Li* et al., 2025) introduces low-rank branches to handle activation outliers, enabling 4-bit quantization without performance degradation. For video generation, ViDiT-Q (Zhao et al., 2025) achieves lossless W8A8 quantization.

Despite these advances, text-to-video generation with quantized DiTs still suffers from two persistent limitations. First, offline calibration for activation pre-scaling in existing video quantization methods (Zhao et al., 2025; Wu et al., 2024; Chen et al., 2025b) imposes heavy computational burdens and struggles to adapt to large activation vatiants. Second, pushing quantization precision below 8 bits for activations (W4A4) triggers significant quality degradation. Our framework overcomes these obstacles through three key innovations presented in Sec. 3.

---

**Algorithm 1 B**ounded-init **G**rid **R**efinement

---

**Input:** $\mathbf{W} \in \mathbb{R}^{n \times m}$, $n \in \mathbb{N}$, $\epsilon \in \mathbb{R}$
**Output:** $\Delta^* \in \mathbb{R}^n$, $\mathbf{z}^* \in \mathbb{Z}^n$
1: $\delta_l = \delta_u = (\max(\mathbf{W}) - \min(\mathbf{W}))/n$
2: $\mathcal{E}^* = \infty, \Delta^* = \mathbf{z}^* = 0$
3: **for** $i = 1, 2, ..., n$ **do**
4: $\quad \mathbf{W}_c = \text{clamp}(\mathbf{W}, \min(\mathbf{W}) + i \cdot \delta_l, \max(\mathbf{W}) - i \cdot \delta_u)$
5: $\quad \Delta^{(0)} = (\max(\mathbf{W}_c) - \min(\mathbf{W}_c))/(2^b - 1), \quad \mathbf{z}^{(0)} = -\lfloor \min(\mathbf{W}_c) \oslash \Delta^{(0)} \rceil$
6: $\quad \mathbf{W}_q^{(0)} = \text{clamp}(\lfloor \mathbf{W} \oslash \Delta^{(0)} \rceil + \mathbf{z}^{(0)}, 0, 2^b - 1)$
7: $\quad \mathcal{E}^{(0)} = \|\mathbf{W} - \Delta^{(0)} \odot (\mathbf{W}_q^{(0)} - \mathbf{z}^{(0)})\|_F, j = 0$
8: $\quad$ **repeat**
9: $\quad\quad \Delta^{(j)} = \langle \mathbf{W}_q^{(j-1)} - \mathbf{z}^{(j-1)}, \mathbf{W} \rangle_{\text{row}} \oslash \langle \mathbf{W}_q^{(j-1)} - \mathbf{z}^{(j-1)}, \mathbf{W}_q^{(j-1)} - \mathbf{z}^{(j-1)} \rangle_{\text{row}}$
10: $\quad\quad \mathbf{z}^{(j)} = \text{clamp}(\overline{\mathbf{W}}_q^{(j-1)} - \overline{\mathbf{W}} \oslash \Delta^{(j)}, 0, 2^b - 1)$
11: $\quad\quad \mathbf{W}_q^{(j)} = \text{clamp}(\lfloor \mathbf{W} \oslash \Delta^{(j)} \rceil + \mathbf{z}^{(j)}, 0, 2^b - 1)$
12: $\quad\quad \mathcal{E}^{(j)} = \|\mathbf{W} - \Delta^{(j)} \odot (\mathbf{W}_q^{(j)} - \mathbf{z}^{(j)})\|_F, j = j + 1$
13: $\quad$ **until** $\mathcal{E}^{(j-1)} - \mathcal{E}^{(j-2)} < \epsilon$
14: $\quad$ **if** $\mathcal{E}^{(j-1)} < \mathcal{E}^*$ **then**
15: $\quad\quad \mathcal{E}^* \leftarrow \mathcal{E}^{(j-1)}, \Delta^* \leftarrow \Delta^{(j-1)}, \mathbf{z}^* \leftarrow \mathbf{z}^{(j-1)}$
16: $\quad$ **end if**
17: **end for**
18: **return** $\Delta^*, \mathbf{z}^*$

---

## 3 METHOD

**Overview.** As shown in Fig. 2, Bounded-init Grid Refinement (BGR) iteratively refines quantization grid upon progressive tightening bounds, which preserves Gaussian-distributed DiT weights with minimal error (Sec. 3.1). Auto-scaling Rotated Quantization (ARQ) eliminates calibration overhead by jointly optimizing rotation and online scaling for activation outliers, which also offers greater flexibility for timestep-dependent scale variants (Sec. 3.2). Additionally, $\delta$-Guided Bit Switching further allocates bit-widths adaptively across timesteps by tracking feature evolution (Sec. 3.3). The synergy of these techniques achieves lower bit-widths (*e.g.*, W4A4 and W4A6) with negligible quality degradation, overcoming limitations of prior quantization approaches.

### 3.1 BOUNDED-INIT GRID REFINEMENT (BGR)

For weight matrix $\mathbf{W} \in \mathbb{R}^{n \times m}$, vanilla MinMax (per-channel) quantization (Jacob et al., 2018) directly adopts the extreme values for range calculation through Quant/DeQuant processes:

$$\text{Quant:} \quad \mathbf{W}_q = \text{clamp}(\lfloor \mathbf{W} \oslash \Delta \rceil + \mathbf{z}, 0, 2^b - 1), \tag{1}$$

$$\text{DeQuant:} \quad \widehat{\mathbf{W}} = \Delta \odot (\mathbf{W}_q - \mathbf{z}), \tag{2}$$

where $\Delta = (\max(\mathbf{W}) - \min(\mathbf{W}))/(2^b - 1) \in \mathbb{R}^n$, $\mathbf{z} = -\lfloor \min(\mathbf{W}) \oslash \Delta \rceil \in \mathbb{Z}^n$ denote the step-size and zero-point for each channel respectively.

This approach is particularly problematic for Gaussian-distributed weights, as fixed ranges amplify quantization errors in two ways: (i) by allocating excessive bins to outlier regions (only 0.3% of parameters), and (ii) by creating suboptimal interval spacing around the zero-mean concentration. To address this, we revisit the optimization objective for low-bit quantization, which is

$$\Delta^*, \mathbf{z}^* = \arg\min_{\Delta, \mathbf{z}} \|\mathbf{W} - \Delta \odot (\mathbf{W}_q - \mathbf{z})\|_F. \tag{3}$$

Notably, the heuristic solution from MinMax algorithm provides an initialization for this optimization problem. As post-training quantization aims to avoid computationally expensive process (*e.g.*, gradient descent), we seek a closed-form solution for the quantization objective. Assuming we have obtained the initial state of $\Delta, \mathbf{z}$, and $\mathbf{W}_q$, our goal is to refine one or all of them to reduce quantization error. Refining them simultaneously is highly challenging, instead, we can refine one parameter while fixing the other two. For instance, we first fix $\mathbf{z}$ and $\mathbf{W}$, then derive a better solution

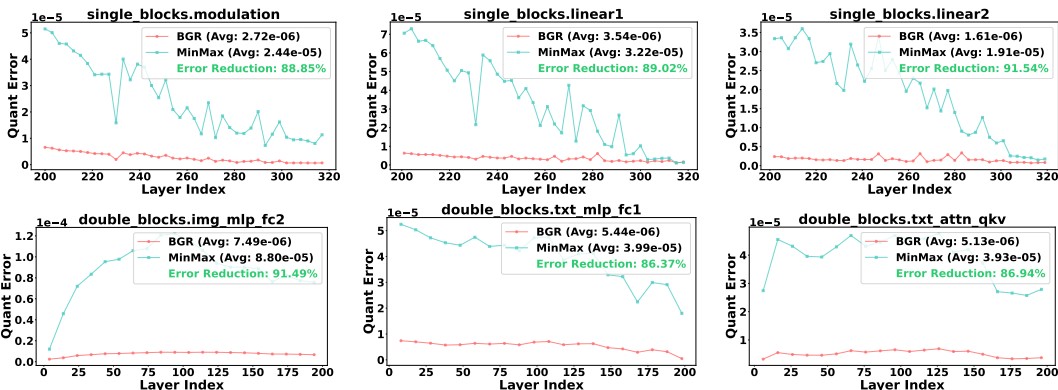

Figure 3: Quantization error comparison: MinMax (Jacob et al., 2018) vs. BGR across layers on HunyuanVideo (Kong et al., 2024).

for $\Delta$ by solving the least squares problem for Eq. 3:

$$\Delta' = \langle \mathbf{W}_q - \mathbf{z}, \mathbf{W} \rangle_{\text{row}} \oslash \langle \mathbf{W}_q - \mathbf{z}, \mathbf{W}_q - \mathbf{z} \rangle_{\text{row}}, \tag{4}$$

where $\langle \cdot, \cdot \rangle_{\text{row}}$ denotes the row-wise inner product, with details provided in the supplementary file.

Once $\Delta'$ is refined, we further refine $\mathbf{z}$ with $\Delta'$ and $\mathbf{W}_q$ fixed. However, $\mathbf{z}$ is restricted to integers and this discreteness precludes a closed-form derivative solution. We thus first relax the range of $\mathbf{z}$ and then apply rounding:

$$\mathbf{z}' = \text{clamp}(\overline{\mathbf{W}}_q - \overline{\mathbf{W}} \oslash \Delta', 0, 2^b - 1). \tag{5}$$

After refining $\Delta'$ and $\mathbf{z}'$, $\mathbf{W}_q$ is also updated under the new quantization grid:

$$\mathbf{W}'_q = \text{clamp}(\lfloor \mathbf{W} \oslash \Delta' \rceil + \mathbf{z}', 0, 2^b - 1), \tag{6}$$

notably, this refinement of $\mathbf{W}_q$ aligns with its definition in Eq. 1.

This sequential refinement procedure can be further extended to multiple rounds and combined with various initialization algorithms. Beyond MinMax, search bound methods (Liu et al., 2024b; Shao et al., 2023) also serve as valid initial strategies, which exclude outliers by progressively clipping full-precision weights:

$$\mathbf{W}_c = \text{clamp}(\mathbf{W}, \min(\mathbf{W}) + \delta_l, \max(\mathbf{W}) - \delta_u), \tag{7}$$

$$\Delta_c = (\max(\mathbf{W}_c) - \min(\mathbf{W}_c))/(2^b - 1), \quad \mathbf{z}_c = -\lfloor \min(\mathbf{W}_c) \oslash \Delta_c \rceil, \tag{8}$$

where $\delta_l$ and $\delta_u$ denote the shrink step sizes for lower and upper bounds, respectively. Since search bound algorithm provides a better initial state for quantization step-size and zero-point, our grid refinement algorithm is built on this initialization (see Alg. 1).

Our experiments in Fig. 3 quantitatively show a substantial **86%** reduction in quantization error, effectively preserving the critical parameters within high-density regions. These gains incur no additional inference-time computational overhead, making BGR more accurate than previous approaches.

## 3.2 AUTO-SCALING ROTATED QUANTIZATION (ARQ)

Activation quantization in Diffusion Transformers (DiTs) poses two fundamental challenges: **First**, the dynamic variation of activations across denoising timesteps renders offline scaling factor calibration ineffective. **Second**, traditional rotation-based methods incur substantial online computational overhead while risking the amplification of quantization errors via value redistribution. Notably, existing approaches fail to address these two challenges comprehensively.

Pre-scaling methods (Xiao et al., 2023; Zhao et al., 2025; Wu et al., 2024) introduce a channel-wise mask $\mathbf{s} \in \mathbb{R}^C$ to alleviate quantization difficulty by transferring it from activations to weights:

$$\mathbf{Y} = (\mathbf{X}\text{diag}(\mathbf{s})^{-1}) \cdot ((\text{diag}(\mathbf{s})\mathbf{W})) = \tilde{\mathbf{X}} \cdot \tilde{\mathbf{W}}, \quad s_i = \max(|\mathbf{X}_i|)^\alpha / \max(|\mathbf{W}_i|)^{1-\alpha}, \tag{9}$$

where $\alpha$ is a hyperparameter that controls the degree of difficulty mitigation. Moreover, these methods typically require a calibration dataset to compute scaling factors offline. When applied to DiT models (which involve 50 denoising steps (Xiao et al., 2023; Zhao et al., 2025)), the calibration set often fails to capture the full dynamic range of activation distributions across all timesteps, introducing substantial quantization errors and potential bias.

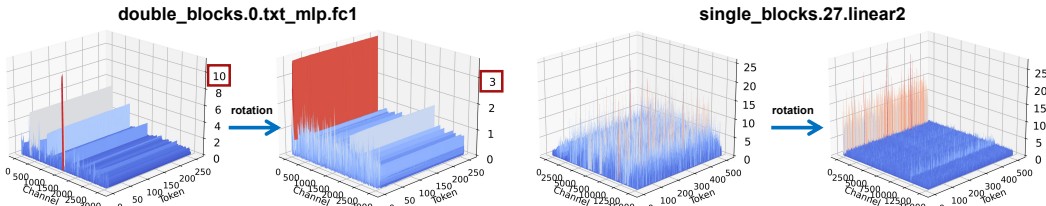

Figure 4: Visualization of activation distribution before and after rotation.

Rotation-based methods (Ashkboos et al., 2024; Liu et al., 2025; Zhao et al., 2025) employ an orthogonal rotation matrix $\mathbf{Q}$ satisfying $\mathbf{Q}\mathbf{Q}^\top = \mathbf{I}$ and $|\mathbf{Q}| = \mathbf{I}$. While multiplying by $\mathbf{Q}$ can effectively suppress large outliers, the rotation operation may inadvertently amplify certain activation values. This risks introducing new quantization errors in the transformed space, as shown in Fig. 4.

To address these challenges, we propose **Auto-scaling Rotated Quantization (ARQ)** for activation quantization. Our approach combines the strengths of both rotation-based and scaling-based methods while overcoming their individual limitations. We first incorporate Hadamard matrix multiplication (Tseng et al., 2024) on both sides of activations and weights to preserve computational invariance: $\mathbf{Y} = \mathbf{X}\mathbf{W}^\top = (\mathbf{X}\mathbf{H})(\mathbf{H}^\top\mathbf{W})$. Hadamard rotation is computed by fast Hadamard transform, which introduces marginal overhead to the inference latency. Subsequently, per-channel scaling factors are computed online and applied directly to activations (rather than being transferred to weights):

$$\widehat{\mathbf{X}} = \mathcal{Q}(\mathbf{X}\mathbf{H}\mathbf{\Lambda}^{-1}), \quad \widehat{\mathbf{W}} = \mathcal{BGR}(\mathbf{W}\mathbf{H}), \quad \mathbf{Y} = \widehat{\mathbf{X}}\mathbf{\Lambda}\widehat{\mathbf{W}}^\top, \tag{10}$$

where $\tilde{\mathbf{X}} = \mathbf{X}\mathbf{H}$, $s_j = \|\tilde{\mathbf{X}}_j\|_\infty = \max_i(|\tilde{x}_{i,j}|)$, and $\mathbf{\Lambda} = \mathrm{diag}(s_1, \ldots, s_c)$.

We analyze how ARQ addresses the massive outliers and channel-wise outliers inherent in DiT activations. For massive outliers, rotation redistributes these extreme values across multiple channels. For channel-wise outliers, the post-rotation online scaling reinforces channel-wise consistency and mitigates the limitation of rotation shown in Fig. 4. Furthermore, our online scaling strategy eliminates reliance on offline calibration datasets. For DiT models, where activation distributions vary drastically across denoising steps, this runtime adaptation ensures optimal quantization parameters at every timestep, sustaining generation quality throughout the entire diffusion process.

In practice, ARQ is implemented under a hardware-aligned constraint that ensures full compatibility with low-bit Tensor Cores. While eq. (10) presents a per-channel scaling formulation for theoretical completeness, our deployed kernels adopt a block-wise scaling variant aligned with Tensor Core's GEMM granularity (more details in supplementary file). All latency and accuracy results in Sec. 4 are derived under this hardware-aligned, block-wise ARQ configuration.

### 3.3 $\delta$-GUIDED BIT SWITCHING ($\delta$-GBS)

The denoising process in video DiTs exhibits non-uniform feature evolution across timesteps. Uniform quantization wastes computational resources on these redundant timesteps while potentially compromising quality during critical transformation phases. Existing adaptive quantization methods either depend on expensive calibration analysis (Zhao et al., 2025) or adopt static timestep segmentation (Wu et al., 2024). Neither of them can capture dynamic feature changes with respect to different input prompts in video generation.

To further enhance video generation quality, we propose $\delta$-Guided Bit Switching, tailored to input characteristics. Prior work (Kahatapitiya et al., 2024; Liu et al., 2024a; Wu et al., 2025; Junhyuk So & Park, 2024; Ma et al., 2024) found that denoising process of DiTs contains redundant timesteps, where latent feature changes are marginal. For these redundant timesteps, we can apply lower bit-width quantization across the entire model. In contrast, we use higher precision for critical timesteps with significant feature transformations. Specifically, our mixed-precision mechanism operates as follows:

$$B_{t_i} = \begin{cases} b_{\text{low}} & \sum_{t=t_p}^{t_{i-1}} \mathcal{L}_1(\mathcal{F}, t) < \delta \\ b_{\text{high}} & \sum_{t=t_p}^{t_{i-1}} \mathcal{L}_1(\mathcal{F}, t) \geq \delta \end{cases}, \quad \mathcal{L}_1(\mathcal{F}, t) = \frac{\|\mathcal{F}_t - \mathcal{F}_{t-1}\|_1}{\|\mathcal{F}_{t-1}\|_1}, \tag{11}$$

where $t_p$ denotes the most recent timestep at which cumulative error tracking is reset, and $\mathcal{F}_t$ represents the model's output features at timestep $t$. Our algorithm works by continuously monitoring normalized L1 distances between successive outputs. When cumulative feature changes remain below the threshold $\delta$, we apply $b_{\text{low}}$-bit quantization, indicating marginal feature evolution can tolerate lower precision. Once the accumulated error exceeds $\delta$, we switch to $b_{\text{high}}$-bit quantization to preserve critical details, while simultaneously resetting the cumulative error counter to zero. This adaptive

Table 1: Performance comparison of various quantization methods on VBench (Huang et al., 2024).

| Method | Bit-width (W/A) | Aesthetic Quality | Imaging Quality | Overall Consist. | Scene Consist. | BG. Consist. | Subject. Consist. | Dynamic Degree | Motion Smooth. |
|---|---|---|---|---|---|---|---|---|---|
| HunyuanVideo (Kong et al., 2024) | 16/16 | 62.53 | 64.78 | 25.86 | 42.81 | 97.01 | 96.05 | 51.39 | 99.30 |
| MinMax (Jacob et al., 2018) | 4/8 | 59.44 | 60.62 | 25.78 | 36.41 | 97.61 | 95.83 | 52.78 | 98.89 |
| SmoothQuant (Xiao et al., 2023) | 4/8 | 60.50 | 64.47 | 25.56 | 28.85 | 97.72 | 96.29 | 51.39 | 99.05 |
| Quarot (Ashkboos et al., 2024) | 4/8 | 58.80 | 56.86 | 25.33 | 34.30 | 98.10 | 95.72 | 55.56 | 99.03 |
| ViDiT-Q (Zhao et al., 2025) | 4/8 | 57.01 | 59.74 | 24.77 | 27.11 | 97.37 | 95.16 | 48.61 | 99.06 |
| $\mathcal{DVD}$-Quant | 4/6 | 62.27 | 64.22 | 25.83 | 33.07 | 97.89 | 96.57 | 58.33 | 99.05 |
| MinMax (Jacob et al., 2018) | 4/4 | 24.20 | 24.78 | 4.27 | 0.00 | 98.05 | 96.27 | 0.00 | 99.03 |
| SmoothQuant (Xiao et al., 2023) | 4/4 | 48.41 | 59.46 | 21.09 | 7.84 | 96.72 | 94.97 | 1.39 | 98.79 |
| Quarot (Ashkboos et al., 2024) | 4/4 | 44.85 | 54.30 | 17.33 | 0.94 | 97.69 | 92.64 | 87.5 | 92.22 |
| ViDiT-Q (Zhao et al., 2025) | 4/4 | 45.36 | 40.10 | 19.66 | 7.85 | 97.19 | 97.29 | 0.00 | 99.43 |
| $\mathcal{DVD}$-Quant | 4/4 | 61.96 | 61.82 | 25.68 | 29.94 | 97.82 | 96.61 | 56.94 | 99.15 |

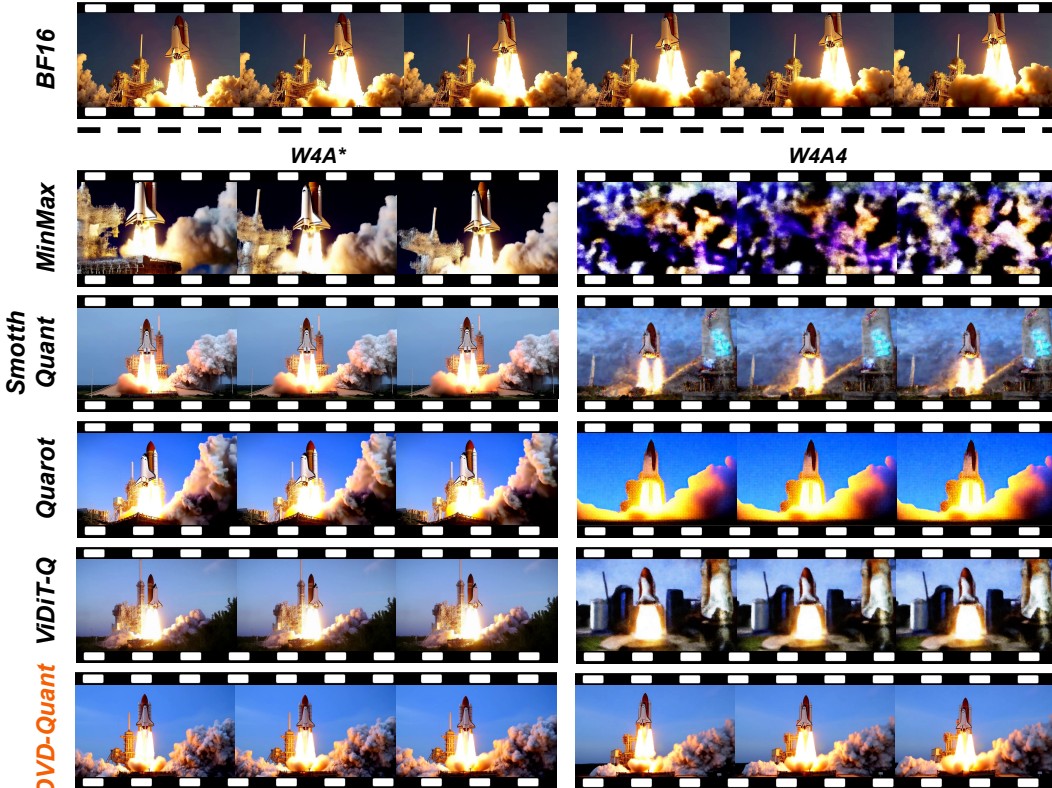

Figure 5: Visual comparisons between $\mathcal{DVD}$-Quant and BF16 baseline (Kong et al., 2024), alongside with quantization methods: MinMax (Jacob et al., 2018), SmoothQuant (Xiao et al., 2023), Quarot (Ashkboos et al., 2024) and ViDiT-Q (Zhao et al., 2025) on HunyuanVideo. * indicates 8 for baselines (W4A8) and 6 for $\mathcal{DVD}$-Quant (W4A6, mixed-precision).

approach optimizes bit allocation while incurring negligible additional inference overhead, with the threshold $\delta$ acting as a hyperparameter to balance performance and efficiency.

# 4 EXPERIMENTS

## 4.1 SETUP

**Video Generation Evaluation Settings:** We apply $\mathcal{DVD}$-Quant to HunyuanVideo (Kong et al., 2024) and generate videos using a 50-step flow matching scheduler (Lipman et al., 2023) with embedded CFG (Ho & Salimans, 2022) scale 6.0 and flow shift factor 7.0. We also apply $\mathcal{DVD}$-Quant to Wan2.1 (Wan et al., 2025), detailed configurations and results are shown in the supplementary file. For evaluation, we adpot the benchmark suite provided by VBench (Huang et al., 2024) to comprehensively assess the quality of the generated videos. Specifically, we evaluate the model across eight major dimensions (Ren et al., 2024) that reflect key aspects of video generation. These metrics are designed to align closely with human perception, ensuring a reliable and standardized

Table 3: Ablation studies of BGR and ARQ under W4A6 and W4A4 quantizations.

| Methods | | Bit Width | Aesthetic | Imaging | Subject | Motion | BG. |
|---|---|---|---|---|---|---|---|
| BGR | ARQ | (W/A) | Quality | Quality | Consist. | Smooth. | Consist. |
| | ✓ | W4A6 | 58.15 | 58.68 | 98.04 | 98.49 | 98.21 |
| ✓ | | W4A6 | 57.85 | 57.72 | 98.23 | 97.86 | 98.10 |
| ✓ | ✓ | W4A6 | 60.46 | 61.93 | 98.91 | 98.95 | 98.40 |
| | ✓ | W4A4 | 53.95 | 52.67 | 97.92 | 98.71 | 97.89 |
| ✓ | | W4A4 | 43.26 | 58.31 | 95.36 | 96.08 | 97.35 |
| ✓ | ✓ | W4A4 | 59.57 | 58.93 | 98.67 | 99.00 | 98.47 |

assessment of model performance. We compare $\mathcal{DVD}$-Quant with MinMax (Jacob et al., 2018), SmoothQuant (Xiao et al., 2023), Quarot (Ashkboos et al., 2024), and the current SOTA video PTQ method ViDiT-Q (Zhao et al., 2025). All experiments are conducted on a single RTX4090 GPU.

## 4.2 MAIN RESULTS

**VBench Quantitative Comparison.** As shown in Tab. 1, $\mathcal{DVD}$-Quant achieves significant improvements over several state-of-the-art quantization methods under two configurations. (1) **High Precision Regime:** When weights are set to 4 bits, existing methods require 8-bit activations to maintain basic functionality, while our configuration demonstrates superior performance with 25% lower activation precision. Notably, our W4A6 mixed-precision configuration nearly matches the BF16 HunyuanVideo model while significantly outperforming all W4A8 baselines across most metrics. (2) **Low Precision Challenge:** In the extremely challenging W4A4 setting where all baseline methods either fail completely or suffer severe degradation, $\mathcal{DVD}$-Quant maintains remarkable stability. For example, we achieve 58.94 Aesthetic Quality and 60.38 Imaging Quality, outperforming the best W4A4 baseline by +10.53 in Aesthetic Quality. Besides, our method preserves temporal dynamics while maintaining high motion smoothness, whereas prior works either degrade visually or fail to generate coherent motion.

**Qualitative Comparison.** In W4A8 settings, while baseline methods preserve only coarse outlines, they still suffer from significant loss of fine details. As illustrated in Fig. 5, ViDiT-Q's W4A8 outputs show washed-out textures in elements like launch towers. In contrast, our method consistently recovers these subtle features, demonstrating robust performance. The advantage becomes more evident in more challenging W4A4 settings, where conventional methods fail completely, generating either incoherent noise patterns or severely distorted outputs. Our method maintains remarkable visual coherence with marginal quality degradation compared to the BF16 baseline.

## 4.3 ABLATION STUDY

**BGR and ARQ.** We validate the effectiveness of BGR and ARQ in Tab. 3. The full model with both BGR and ARQ achieves optimal performance across all metrics, demonstrating their synergistic effects. Removing only BGR leads to significant drops in VBench scores, indicating the critical role of progressively narrowing the weight quantization bound to preserve perceptual fidelity. Conversely, disabling ARQ components results in comparable declines, highlighting the importance of dynamically adjusting quantization scales during inference. Both BGR and ARQ maintain their effectiveness irrespective of $\delta$-GBS.

**Comparison with other mixed-precision strategies.** In the experiment, we set $b_{low} = 4$ bit and $b_{high} = 8$ bit in Eq. 11. Our method achieves an average bit width of 6 across 50 timesteps. We compare the proposed $\delta$-Guided Bit Switching with other fixed-pattern mixed-precision strategies. Specifically, we compare with four representative approaches: (1) **Static Temporal Partitioning (STP):** allocate 4-bit precision for the first 25 timesteps and 8-bit precision for the subsequent 25 timesteps; (2) **Inverse Temporal Partitioning (ITP):** reverse the bitwidth allocation order of STP with 8-bit initially and 4-bit later; (3) **Alternating Bitwidth Switching (ABS):** dynamically alternate between 4-bit and 8-bit precision at each timestep; (4) **Stochastic Bitwidth Allocation (SBA):** randomly assign 4-bit or 8-bit precision to each timestep while maintaining the average bit number 25. As shown in Tab. 2, $\delta$-Guided Bit Switching surpasses other static mixed-precision strategies in imaging quality. This error-driven dynamic decision mechanism effectively resolves the challenge of bit allocation in conventional mixed-precision strategies.

Table 2: Performance comparison of Different mixed-precision Strategies.

| Method | $\delta$-GBS | STP | ITP | ABS | SBA |
|---|---|---|---|---|---|
| Imaging Quality | 61.93 | 61.33 | 61.40 | 61.03 | 61.26 |

Table 4: Performance and speedup of $\mathcal{DVD}$-Quant + TeaCache (Liu et al., 2024a).

| Method | Bit-width (W/A) | Aesthetic Quality | Imaging Quality | BG. Consist. | Subject. Consist. | Motion Smooth. | Speedup |
|---|---|---|---|---|---|---|---|
| $\mathcal{DVD}$-Quant+TeaCache | W4A8 | 61.39 | 62.96 | 98.24 | 98.73 | 98.75 | 4.01× |
| | W4A4 | 58.78 | 58.20 | 98.43 | 98.68 | 98.92 | 4.85× |

**Threshold of $\delta$-Guided Bit Switching.** In our mixed-precision strategy, the threshold $\delta$ determines when to switch bit-width during inference. Empirically, a lower $\delta$ leads to more time steps using high-bit quantization, improving visual quality at the cost of increased bit-width. This presents a trade-off between performance and resource usage. Thus, we conduct a series of experiments with different thresholds. As shown in Fig. 6, our experiments demonstrate that as $\delta$ increases, the average bit-width decreases, leading to a gradual reduction in imaging quality. Notably, a key advantage of $\delta$-GBS over existing methods is its smooth and continuous bit-width adaptation: when $\delta \to 0$, the model consistently uses W4A8; when $\delta \to \infty$, it reduces to W4A4. Critically, for intermediate $\delta$ values, $\delta$-GBS dynamically interpolates between 4-bit and 8-bit at the activation level, ensuring a continuous precision transition based on input characteristics. This smooth adaptation is in stark contrast to conventional approaches, which typically switch abruptly between discrete bit-widths, causing instability in output quality.

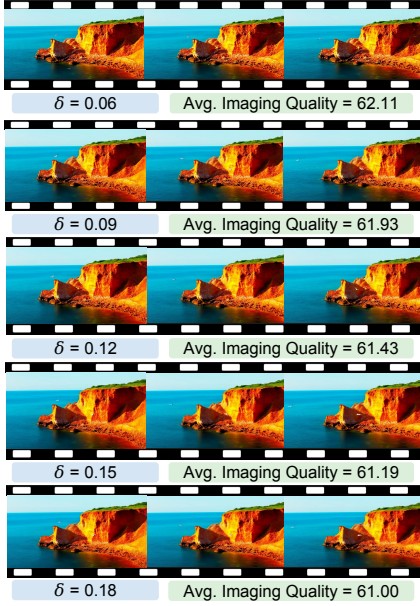

$\delta = 0.06$ — Avg. Imaging Quality = 62.11
$\delta = 0.09$ — Avg. Imaging Quality = 61.93
$\delta = 0.12$ — Avg. Imaging Quality = 61.43
$\delta = 0.15$ — Avg. Imaging Quality = 61.19
$\delta = 0.18$ — Avg. Imaging Quality = 61.00

Figure 6: Comparison of different $\delta$.

### 4.4 MEMORY AND LATENCY

$\mathcal{DVD}$-Quant achieves significant improvements in both memory efficiency and inference speed. As shown in Tab. 5, compared to the BF16 baseline (16/16 bit-width), our W4A8 configuration reduces memory usage by 3.68× while accelerating latency by 1.75×. More aggressive quantization (W4A4) maintains the same memory savings but further boosts latency speedup to 2.12×. Notably, W4A6 means roughly half the denoising steps use W4A8 and the other half use W4A4. This allows direct use of the widely adopted W4A4 and W4A8 GEMM kernels, eliminating the need to design dedicated mixed-precision kernels.

Table 5: Memory saving and latency speedup of $\mathcal{DVD}$-Quant on HunyuanVideo.

| Bit-width (W/A) | Memory Opt. | Latency Opt. |
|---|---|---|
| 16/16 | 1.00× | 1.00× |
| 4/8 (ours) | 3.68× | 1.75× |
| 4/6 (ours) | 3.68× | 1.93× |
| 4/4 (ours) | 3.68× | 2.12× |

### 4.5 INTEGRATE WITH CACHE MECHANISM

To demonstrate the orthogonal compatibility of $\mathcal{DVD}$-Quant with other compression paradigms, we integrate it with one of the SOTA cache methods TeaCache (Liu et al., 2024a). As illustrated in Tab. 4, $\mathcal{DVD}$-Quant maintains its compression efficiency alongside TeaCache (Liu et al., 2024a), achieving additive speedup (up to **4.85×**) without degrading key video generation metrics. This validates the practicality of $\mathcal{DVD}$-Quant in resource constrained scenarios where multiple optimization dimensions could be jointly addressed.

## 5 CONCLUSION

In this work, we propose $\mathcal{DVD}$-Quant, a comprehensive quantization framework including three main innovations. The Bounded-init Grid Refinement (BGR) automatically adapts to weight distributions through iterative grid refinement upon bound tightening, while Auto-scaling Rotated Quantization (ARQ) eliminates calibration dependencies through online rotation and scaling. The $\delta$-Guided Bit Switching further optimizes computational efficiency through content-aware precision allocation. Extensive experiments demonstrate that $\mathcal{DVD}$-Quant achieves 2× speedup on advanced DiT models while maintaining visual quality, becoming the first framework to successfully enable W4A4 post-training quantization for video generation tasks.

## ACKNOWLEDGMENTS

This work is supported by the National Natural Science Foundation of China (62501386), Shanghai Municipal Science and Technology Major Project (2021SHZDZX0102), and the Fundamental Research Funds for the Central Universities. This work is also sponsored by CCF-Tencent Rhino-Bird Open Research Fund.

## ETHICS STATEMENT

The research conducted in the paper conforms, in every respect, with the ICLR Code of Ethics.

## REPRODUCIBILITY STATEMENT

We have provided implementation details in Sec. 4. We will also release all the code and models.

## LLM USAGE STATEMENT

Large Language Models (LLMs) were used solely for polishing writing. They did not contribute to the research content or scientific findings of this work.

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
