# 💿 $\mathcal{DVD}$-Quant: Data-free Video Diffusion Transformers Quantization

**Zhiteng Li**[1*], **Hanxuan Li**[2*†], **Junyi Wu**[1], **Kai Liu**[1], **Haotong Qin**[3], **Linghe Kong**[1],
**Guihai Chen**[1], **Yulun Zhang**[1], **Xiaokang Yang**[1‡]
[1]Shanghai Jiao Tong University, [2]Zhejiang University, [3]ETH Zürich

## 1 More Details of Bounded-init Grid Refinement

Here, we provide the detailed derivation of Bounded-init Grid Refinement (BGR). Start from the per-channel quantization error:

$$\mathcal{E} = \|\mathbf{W} - \Delta \odot (\mathbf{W}_q - \mathbf{z})\|_F$$
$$= \sum_{i,j}(W_{i,j} - \Delta_i \cdot ((W_q)_{i,j} - z_i), \tag{1}$$

where $\Delta = (\Delta_1, \Delta_2, \ldots, \Delta_n)^\top$ and $\mathbf{z} = (z_1, z_2, \ldots, z_n)^\top$ denote the per-channel step-size and zero-point, respectively.

Assuming $\mathbf{z}$ and $\mathbf{W}_q$ are fixed, and noting that the channels are independent ($\Delta_k$ only affect the $k$-th channel elements), we first take the derivative with respect to $\Delta_k$:

$$\frac{\partial \mathcal{E}}{\partial \Delta_k} = 2\sum_j(W_{k,j} - \Delta_k \cdot ((W_q)_{k,j} - z_k)), \tag{2}$$

setting this derivative to zero yields:

$$\sum_j W_{k,j} \cdot ((W_q)_{k,j} - z_k) = \Delta_k \sum_j((W_q)_{k,j} - z_k)^2, \tag{3}$$

$$\Rightarrow \Delta_k = \frac{\sum_j W_{k,j} \cdot ((W_q)_{k,j} - z_k)}{\sum_j((W_q)_{k,j} - z_k)^2}, \tag{4}$$

combining results across all channels gives the vectorized form:

$$\Delta = \langle \mathbf{W}_q - \mathbf{z}, \mathbf{W} \rangle_{\text{row}} \oslash \langle \mathbf{W}_q - \mathbf{z}, \mathbf{W}_q - \mathbf{z} \rangle_{\text{row}}. \tag{5}$$

To derive update formula of $\mathbf{z}$, we assume $\Delta$ and $\mathbf{W}$ are fixed. Since $\mathbf{z}$ is constrained to integers, finding a closed-form solution is non-trivial.

We aim to find $\mathbf{z}$ that minimizes the quantization error. Rearranging the approximation and relaxing the integer constraint of $\mathbf{z}$:

$$\Delta \odot (\mathbf{W}_q - \mathbf{z}) \approx \mathbf{W}, \tag{6}$$

focusing on the $k$-th channel:

$$\sum_j \Delta_k \cdot ((W_q)_{k,j} - z_k) \approx \sum_j W_{k,j}, \tag{7}$$

$$\Rightarrow z_k \approx \frac{1}{m}(\sum_j(W_q)_{k,j} - \frac{\sum_j W_{k,j}}{\Delta_k}), \tag{8}$$

rounding $z_k$ to the nearest integer and combining across channels (with clamping to ensure valid $b$-bit representation):

$$\mathbf{z} = \text{clamp}\left(\overline{\mathbf{W}}_q - \overline{\mathbf{W}} \oslash \Delta, 0, 2^b - 1\right). \tag{9}$$

---
*Equal contribution
†Work done during an internship at Shanghai Jiao Tong University
‡Corresponding author: Xiaokang Yang, xkyang@sjtu.edu.cn

The final step in each BGR iteration is to re-round $\mathbf{W}_q$ under the updated quantization grid:

$$\mathbf{W}_q = \mathrm{clamp}(\lfloor \mathbf{W} \oslash \Delta \rceil + \mathbf{z}, 0, 2^b - 1). \tag{10}$$

This step is critical, as subsequent updates of $\Delta$ and $\mathbf{z}$ depend on the refined $\mathbf{W}_q$ for improved accuracy.

## 2  DIFFUSION MODELS

Diffusion models (Ho et al., 2020; Nichol & Dhariwal, 2021; Kingma et al., 2021; Karras et al., 2022; Song et al., 2021; Chen et al., 2023; Salimans & Ho, 2022) have emerged as a powerful class of generative models, drawing inspiration from thermodynamic diffusion processes where particles undergo random motion leading to gradual dispersion. These models operate through two fundamental phases: a forward process that systematically corrupts data with noise, and a reverse process that learns to denoise and reconstruct the original data distribution.

### 2.1  FORWARD DIFFUSION PROCESS

The forward process begins with a clean data sample $x_0 \sim q(x)$ and applies $T$ successive noise injections. At each timestep $t$, the corrupted data is given by:

$$x_t = \sqrt{\bar{\alpha}_t}x_0 + \sqrt{1 - \bar{\alpha}_t}\epsilon_t, \quad \epsilon_t \sim \mathcal{N}(0, I), \tag{11}$$

where $\bar{\alpha}_t$ represents a carefully designed noise schedule that determines the rate of corruption, and $\epsilon_t$ is standard Gaussian noise. This process transforms the data distribution gradually into an isotropic Gaussian distribution by timestep $T$.

### 2.2  REVERSE DENOISING PROCESS

The reverse process learns a parameterized transition kernel $p_\theta(x_{t-1}|x_t)$ that progressively removes noise. This is typically formulated as a Gaussian distribution:

$$p_\theta(x_{t-1}|x_t) = \mathcal{N}\left(x_{t-1}; \mu_\theta(x_t, t), \Sigma_\theta(x_t, t)\right), \tag{12}$$

where $\mu_\theta$ and $\Sigma_\theta$ are neural networks that predict the mean and covariance of the denoised distribution. Advanced implementations often employ variance-preserving or variance-exploding techniques to improve sample quality.

### 2.3  TRAINING OBJECTIVES

Diffusion models are typically trained using variational lower bounds or simplified objectives like:

$$\mathcal{L}(\theta) = \mathbb{E}_{t,x_0,\epsilon}\left[\|\epsilon - \epsilon_\theta(x_t, t)\|^2\right], \tag{13}$$

where $\epsilon_\theta$ predicts the noise component. This formulation enables stable training and has connections to score-based generative modeling.

Currently, neither video diffusion models nor our quantization approach present any foreseeable negative societal consequences.

## 3  ADDITIONAL RESULTS

**Model Configuration.**    For HunyuanVideo (Kong et al., 2024), we generate video clips of 544x690 resolution and 65 frames. For Wan2.1 T2V-1.3B (Wan et al., 2025), we use a 50-step UniPC multistep solve (Zhao et al., 2023) adapted for flow matching scheduler (Lipman et al., 2023) with embedded classifier-free guidance scale of 6.0 and a sample shift factor of 8.0 to generate video clips of 832x480 resolution and 81 frames.

**Quantitative Comparison.**    Due to page limit, we present the VBench performance of Wan2.1 in Tab. 1. As can be observed, $\mathcal{DVD}$-Quant outperforms other quantization methods by a significant

Table 1: Wan2.1 performance comparison of various quantization methods on VBench (Huang et al., 2024).

| Method | Bit-width (W/A) | Aesthetic Quality | Imaging Quality | Overall Consist. | Scene Consist. | BG. Consist. | Subject. Consist. | Dynamic Degree | Motion Smooth. |
|---|---|---|---|---|---|---|---|---|---|
| Wan2.1-1.3B (Wan et al., 2025) | 16/16 | 64.51 | 68.02 | 23.38 | 22.60 | 98.04 | 95.76 | 73.61 | 98.38 |
| MinMax (Jacob et al., 2018) | 4/8 | 57.61 | 63.01 | 23.04 | 15.99 | 96.25 | 94.18 | 59.72 | 97.36 |
| SmoothQuant (Xiao et al., 2023) | 4/8 | 60.15 | 64.98 | 22.35 | 22.46 | 96.48 | 95.65 | 61.11 | 97.99 |
| Quarot (Ashkboos et al., 2024) | 4/8 | 60.16 | 66.05 | 22.29 | 20.20 | 97.15 | 95.44 | 50.00 | 98.36 |
| ViDiT-Q (Zhao et al., 2025) | 4/8 | 56.70 | 62.10 | 6.74 | 20.64 | 96.07 | 94.58 | 47.22 | 97.65 |
| $\mathcal{DVD}$-Quant | 4/6 | 63.17 | 66.89 | 23.37 | 19.04 | 97.74 | 95.66 | 61.12 | 98.26 |
| MinMax (Jacob et al., 2018) | 4/4 | 32.61 | 52.03 | 2.21 | 0.00 | 95.85 | 90.71 | 100.00 | 87.33 |
| SmoothQuant (Xiao et al., 2023) | 4/4 | 30.70 | 46.15 | 3.52 | 0.02 | 95.13 | 89.73 | 100.00 | 88.28 |
| Quarot (Ashkboos et al., 2024) | 4/4 | 33.34 | 46.28 | 5.84 | 0.00 | 95.67 | 91.41 | 100.00 | 90.40 |
| ViDiT-Q (Zhao et al., 2025) | 4/4 | 32.03 | 52.02 | 2.11 | 0.00 | 95.95 | 90.15 | 100.00 | 87.15 |
| $\mathcal{DVD}$-Quant | 4/4 | 58.94 | 60.38 | 25.62 | 13.80 | 95.32 | 90.93 | 54.17 | 96.37 |

Table 2: HunyuanVideo performance comparison of various quantization methods.

| Method | Bit-width (W/A) | CLIPSIM | CLIP-Temp | VQA-Aesthetic | VQA-Technical | $\Delta FlowScore(\downarrow)$ |
|---|---|---|---|---|---|---|
| HunyuanVideo (Kong et al., 2024) | 16/16 | 0.1850 | 0.9994 | 98.64 | 12.23 | – |
| MinMax (Jacob et al., 2018) | 4/8 | 0.1841 | 0.9991 | 95.49 | 10.34 | 0.625 |
| SmoothQuant (Xiao et al., 2023) | 4/8 | 0.1879 | 0.9994 | 96.95 | 10.86 | 0.529 |
| Quarot (Ashkboos et al., 2024) | 4/8 | 0.1860 | 0.9986 | 94.63 | 8.59 | 0.423 |
| ViDiT-Q (Zhao et al., 2025) | 4/8 | 0.1857 | 0.9991 | 94.00 | 10.02 | 0.857 |
| $\mathcal{DVD}$-Quant | 4/6 | 0.1870 | 0.9991 | 98.30 | 11.83 | 0.342 |
| MinMax (Jacob et al., 2018) | 4/4 | 0.1798 | 0.9994 | 8.54 | 1.45 | 1.204 |
| SmoothQuant (Xiao et al., 2023) | 4/4 | 0.1831 | 0.9995 | 55.42 | 6.03 | 1.282 |
| Quarot (Ashkboos et al., 2024) | 4/4 | 0.1766 | 0.9966 | 30.94 | 4.16 | 4.601 |
| ViDiT-Q (Zhao et al., 2025) | 4/4 | 0.1867 | 0.9994 | 39.85 | 4.52 | 1.381 |
| $\mathcal{DVD}$-Quant | 4/4 | 0.1872 | 0.9991 | 97.71 | 10.88 | 0.556 |

margin. Notably, under the W4A4 setting, where most competing methods struggle to deliver valid results, $\mathcal{DVD}$-Quant still maintains reasonable performance. In addition to the VBench (Huang et al., 2024) metrics, we incorporate several representative evaluation criteria: *CLIPSIM* and *CLIP-Temp* (Liu et al., 2024) for assessing text-video alignment and temporal semantic consistency, DOVER's (Wu et al., 2023) video quality assessment(*VQA*) metrics for evaluating generation quality from both aesthetic and technical perspectives, and *Flow-score* for measuring temporal consistency. As shown in Table 2, our method demonstrates superior performance across both quantization scenarios. Our method in the W4A6 configuration surpasses W4A8 baseline approaches. The advantages become more pronounced in the more challenging W4A4 scenario, where the performance gap in video quality metrics between our method and others widens substantially.

**Qualitative Comparison.** As shown in Figures 1 to 3, we provide visual comparisons between our method and other baseline approaches (Jacob et al., 2018; Zhao et al., 2025; Xiao et al., 2023; Ashkboos et al., 2024)under different quantization settings. In the challenging W4A4 scenario, baseline methods fail to generate visually coherent results, typically producing either severe noise artifacts or heavily blurred outputs. While the W4A8 setting allows baseline methods to maintain basic structural outlines, they still exhibit deficiencies in fine-grained details. In contrast, our approach consistently generates high-quality visual content across both quantization configurations, demonstrating superior robustness to bit-width reduction.

**Weight Distribution Analysis.** Figures 6 and 7 present the detailed weight distribution visualization of HunyuanVideo (Kong et al., 2024). The histograms plot weight values on the x-axis against their frequency counts on the y-axis. Our analysis reveals two key characteristics: (1)**Gaussian Dominance.** The weights predominantly follow a Gaussian distribution pattern, and (2)**Long-Tail Phenomenon.** All layers exhibit noticeable long-tail characteristics where rare but extreme weight values persist despite their low occurrence frequency.

**Quantization Error Analysis.** As illustrated in Figures 4 and 5, we present a comprehensive comparison of quantization error distributions between MinMax (Jacob et al., 2018) and our proposed

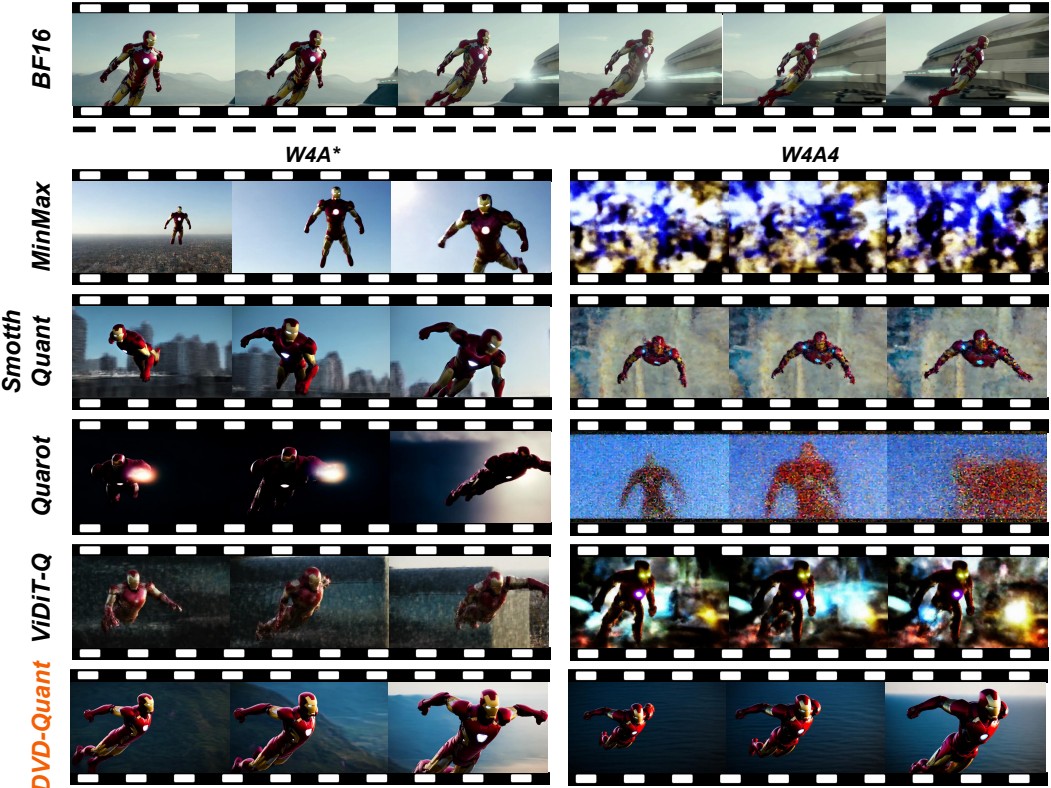

Figure 1: Visual comparisons with prompt: Iron Man flying in the sky. * indicates 8 for baselines (W4A8) and 6 for $\mathcal{DVD}$-Quant (W4A6, mixed-precision).

ARQ across different layers. The visualization reveals that ARQ maintain significantly tighter error clustering around zero compared to MinMax's dispersed outliers, validating our method's superior capability in minmizing quatization error.

## 4 LIMITATIONS AND FUTURE WORKS

Our approach is primarily evaluated on standard benchmarks, and its behavior under extreme or unconventional settings remains an open question. We can integrate more auxiliary components like sparse attention (Zhang et al., 2025) to achieve a better speedup ratio. Additionally, we plan to implement more experiments on edge devices in the future.

## 5 SOCIAL IMPACTS

Video generation models (Kong et al., 2024) have become promising for applications like film production, virtual reality, and automated content creation. However, existing low-bit quantization methods either fail to maintain generation quality or cannot be effectively applied to video diffusion models. Our work breaks this barrier by introducing one of the first W4A4 quantization frameworks that achieve near-lossless video generation while enabling real-time deployment on edge devices. This advancement paves the way for high-quality, efficient video synthesis in resource-constrained scenarios.

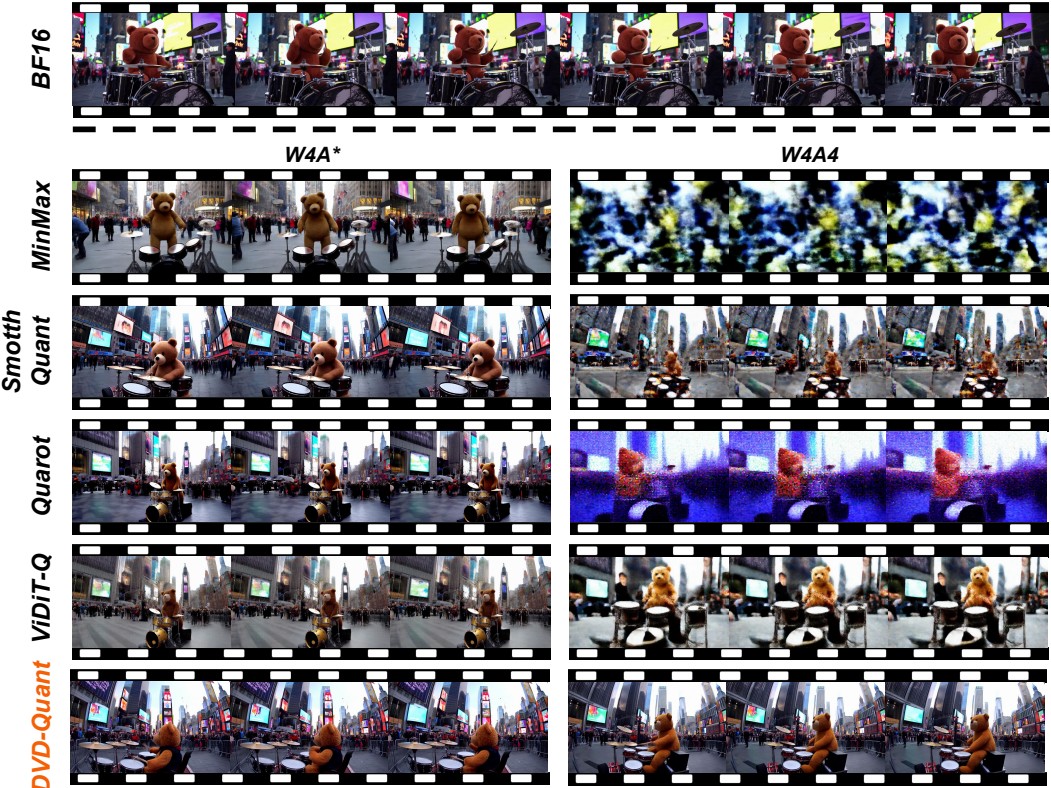

Figure 2: Visual comparisons with prompt: A teddy bear is playing drum kit in NYC Times Square. * indicates 8 for baselines (W4A8) and 6 for $\mathcal{DVD}$-Quant (W4A6, mixed-precision).

## 6 MORE DISCUSSIONS ON ROTATION-BASED QUANTIZATION

Beyond QuaRot (Ashkboos et al., 2024), several recent methods also exploit rotations to mitigate activation outliers in LLM quantization, including ResQ, RoSTE, and DuQuant (Lin et al., 2024; Wei et al., 2025; Saxena et al., 2024).

ResQ (Saxena et al., 2024) adopts a data-driven post-training strategy: it requires a calibration dataset to perform Principal Component Analysis (PCA) and then applies random rotations within the learned principal subspaces to suppress outliers. As a result, its effectiveness is tied to the quality and domain coverage of the calibration data, and it incurs non-trivial costs for data collection and preprocessing. By contrast, DVD-Quant is fully data-free: we employ analytic Hadamard-based rotations that do not depend on calibration samples or prior knowledge of the input distribution, which avoids potential overfitting to a specific calibration set and eliminates the overhead of preparing calibration data.

RoSTE (Wei et al., 2025), in turn, is formulated as a quantization-aware training (QAT) method integrated into supervised fine-tuning (SFT). It jointly optimizes model weights and rotation configurations via computationally intensive retraining, and treats the rotation selection as a bilevel optimization problem that searches, for each layer, between the identity and Walsh–Hadamard transforms based on training data. In contrast, DVD-Quant is a pure post-training quantization (PTQ) method: we rely on fixed, optimization-free rotation matrices and do not require any gradient-based search or access to full fine-tuning datasets, which leads to a much simpler and cheaper quantization pipeline than RoSTE's search-based design.

DuQuant (Lin et al., 2024) also introduces rotation into the quantization process but differs from our approach in both the order of operations and computational complexity. DuQuant first performs activation-dependent scaling and then constructs input-conditioned rotation matrices, making the rotation explicitly dependent on the current activations and involving relatively heavy matrix construction at inference time. In contrast, our DVD-Quant (ARQ) variant applies a fixed rotation *before* scaling and adjusts only the scaling coefficients through a lightweight auto-scaling step based on

Table 3: Symmetric vs. asymmetric quantization comparison at W4A4 precision.

| Method | Bit-width (W/A) | Aesthetic Quality | Imaging Quality | Subject Consist. | Motion Smooth | BG Consist. | Speedup |
|---|---|---|---|---|---|---|---|
| ViDiT-Q | W4A4 | 44.18 | 40.40 | 98.10 | 99.37 | 97.80 | - |
| DVD-Quant (asym) | W4A4 | 59.57 | 58.93 | 98.67 | 99.00 | 98.47 | 2.12× |
| DVD-Quant (sym) | W4A4 | 59.23 | 58.20 | 98.27 | 98.99 | 97.91 | 2.39× |

simple activation statistics. This design preserves the benefits of rotation-based outlier suppression while keeping the dynamic computation extremely cheap, making DVD-Quant considerably more deployment-friendly than DuQuant.

# 7 More results on the generalization of BGR

BGR operates as a quantization-agnostic optimization framework, natively supporting both symmetric and asymmetric quantization schemes without architectural bias. The core formulation inherently accommodates symmetric quantization by eliminating zero-point dependencies, causing cross-terms to vanish naturally while incurring zero computational overhead relative to standard symmetric implementations.

While our primary evaluation employs asymmetric quantization to maximize accuracy preservation, a configuration well-supported by hardware-aware optimizers, BGR equally enables hardware-optimal symmetric quantization. As demonstrated in Tab. 3, symmetric quantization achieves near-identical performance to asymmetric variants across all quality metrics, while delivering measurable inference acceleration. This flexibility establishes BGR as a unified solution adaptable to diverse deployment constraints without sacrificing output fidelity.

# 8 More Discussions on Kernel Implementation

In this section, we provide a detailed analysis of the hardware implementation of our proposed method, specifically focusing on the efficiency of asymmetric quantization, the compatibility of Auto-scaling Rotated Quantization (ARQ) with low-bit Tensor Cores, the kernel framework, and the mixed-precision speed hierarchy.

## 8.1 Asymmetric Quantization in BGR

**Compatibility of BGR.** We first emphasize that the Bounded-init Grid Refinement (BGR) framework is agnostic to the quantization scheme. It is compatible with both symmetric and asymmetric quantization and does not intrinsically rely on the latter.

- When applied to symmetric quantization (where zero-points $Z_x = Z_w = 0$), the cross-terms naturally vanish, and there is no additional overhead compared to standard symmetric methods.

- Asymmetric quantization is utilized in our experiments primarily to demonstrate the method's capability in preserving higher accuracy, which is a common trade-off in low-bit scenarios.

**Efficient Implementation via Epilogue Fusion.** For asymmetric configurations, the term involving token sums ($\mathbf{X} \cdot Z_w$) requires careful handling to avoid overhead. We follow the industry-standard "Quantization-Epilogue Fusion" paradigm used in mainstream frameworks (e.g., FBGEMM (Khudia et al., 2021), NVIDIA TensorRT (NVIDIA)). The reduction sum of activations $\mathbf{X}$ is computed via **kernel fusion**, performed on-the-fly while loading data into registers for the main GEMM kernel. Since this operation is typically memory-bound, piggybacking the summation onto the memory load effectively hides the computational latency. Consequently, the measured latency difference between asymmetric and symmetric implementations is negligible ($< 2\%$).

## 8.2 ARQ Compatibility with Low-bit Tensor Cores

Standard fine-grained per-channel activation quantization is often incompatible with INT4 Tensor Cores because unique scaling factors cannot be factored out of the accumulation. Our **Auto-scaling Rotated Quantization (ARQ)** explicitly resolves this bottleneck by combining **Hadamard Rotation** with **Hardware-Aligned Grouping**.

**1. Hardware-Aligned Implementation.** While the theoretical formulation defines scaling per-channel, our efficient implementation aligns these scales with hardware GEMM blocks.

- **Hardware Alignment:** Instead of assigning a unique floating-point scale to every single channel, we enforce that quantization scales are shared within a **computation block** (aligned with the Tensor Core GEMM block size, e.g., 128).
- **Role of Hadamard Rotation:** By applying Hadamard Rotation first, we redistribute massive outliers across channels. This "flattening" effect allows the use of block-shared scales without the precision loss that usually necessitates strictly fine-grained (and slow) per-channel scaling.

**2. Mathematical Formulation.** Due to block alignment, the scale becomes a constant term within the accumulation loop. ARQ introduces an additional block-wise scaling factor $a^{(b)}$. The computation for one block is formulated as:

$$\mathbf{Y}^{(b)} = a^{(b)} \sum_{k=1}^{K^{(b)}} \hat{\mathbf{X}}_k \hat{\mathbf{W}}_k^\top \tag{14}$$

where $\mathbf{Y}^{(b)}$ is the block output, $a^{(b)}$ is the ARQ block-wise scale, and $\hat{\mathbf{X}}_k$, $\hat{\mathbf{W}}_k$ are the INT4 activation and weight tiles processed by the Tensor Core. (Base quantization scales are omitted for simplicity as they follow standard handling).

- **Tensor Core Stage:** The summation term involving $\hat{\mathbf{X}}$ and $\hat{\mathbf{W}}$ is a pure integer-to-integer matrix multiplication, executed at full throughput by INT4 Tensor Cores.
- **Epilogue Stage:** The multiplication by the floating-point scalar $a^{(b)}$ is computationally lightweight and is fused into the GEMM Epilogue, occurring after the heavy accumulation.

**3. Performance Verification.** Experimental results confirm this compatibility. DVD-Quant achieves a **2.12× latency speedup** over the BF16 baseline, confirming the leverage of low-bit Tensor Core throughput.

## 8.3 Kernel Framework and Implementation

**Implementation with NVIDIA CUTLASS.** Our kernel implementation supporting ARQ is built upon the widely adopted **NVIDIA CUTLASS** library.

- **Standard Main Loop:** The compute-intensive matrix multiplication utilizes the standard INT4 Tensor Core MMA pipeline provided by CUTLASS without modification, ensuring peak throughput.
- **Customized Epilogue:** We implemented a custom epilogue to handle the ARQ scaling $a^{(b)}$ and asymmetric corrections efficiently while data resides in registers (as detailed in Section 8.2).

**Elimination of Mixed-Precision Kernels.** Our claim regarding "eliminating the need to design dedicated mixed-precision kernels" refers to the $\delta$-Guided Bit Switching ($\delta$-GBS) mechanism. Since $\delta$-GBS allocates precision at the **timestep level** (i.e., an entire denoising step is either W4A4 or W4A8), we simply switch between invoking a standard W4A4 kernel and a W4A8 kernel. This avoids the complexity and inefficiency of designing intra-operator mixed-precision kernels (e.g., mixing 4-bit and 8-bit channels within a single GEMM).

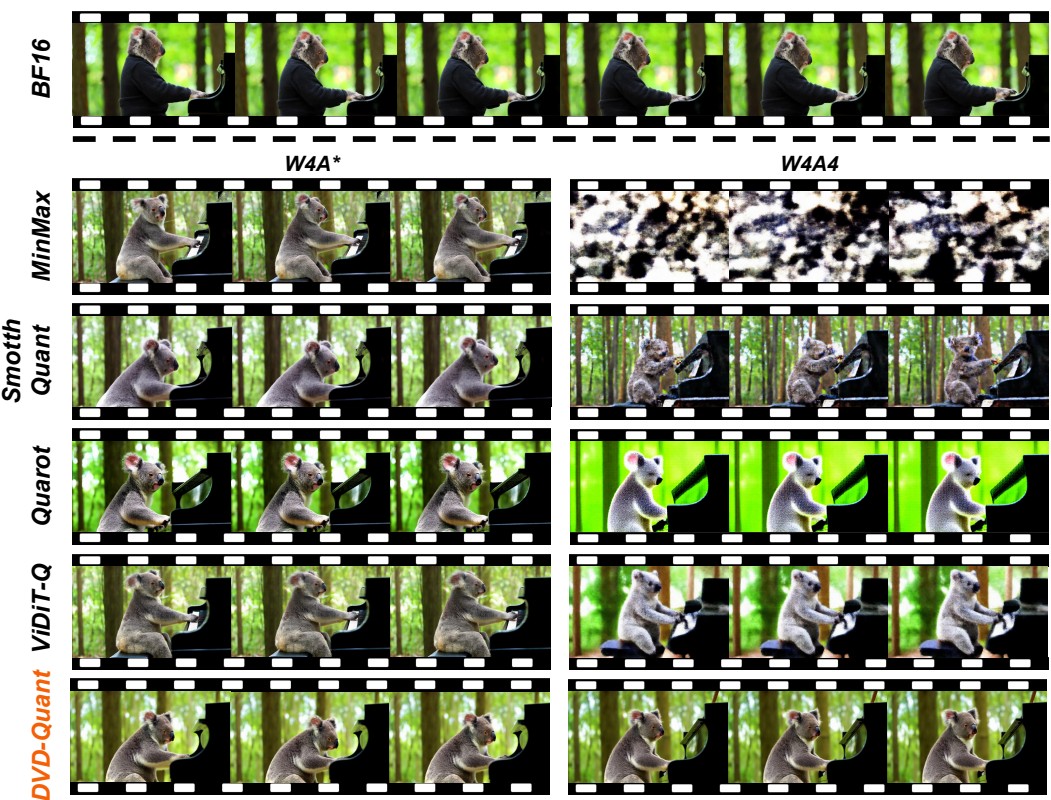

Figure 3: Visual comparisons with prompt: A koala bear playing piano in the forest. * indicates 8 for baselines (W4A8) and 6 for $\mathcal{DVD}$-Quant (W4A6, mixed-precision).

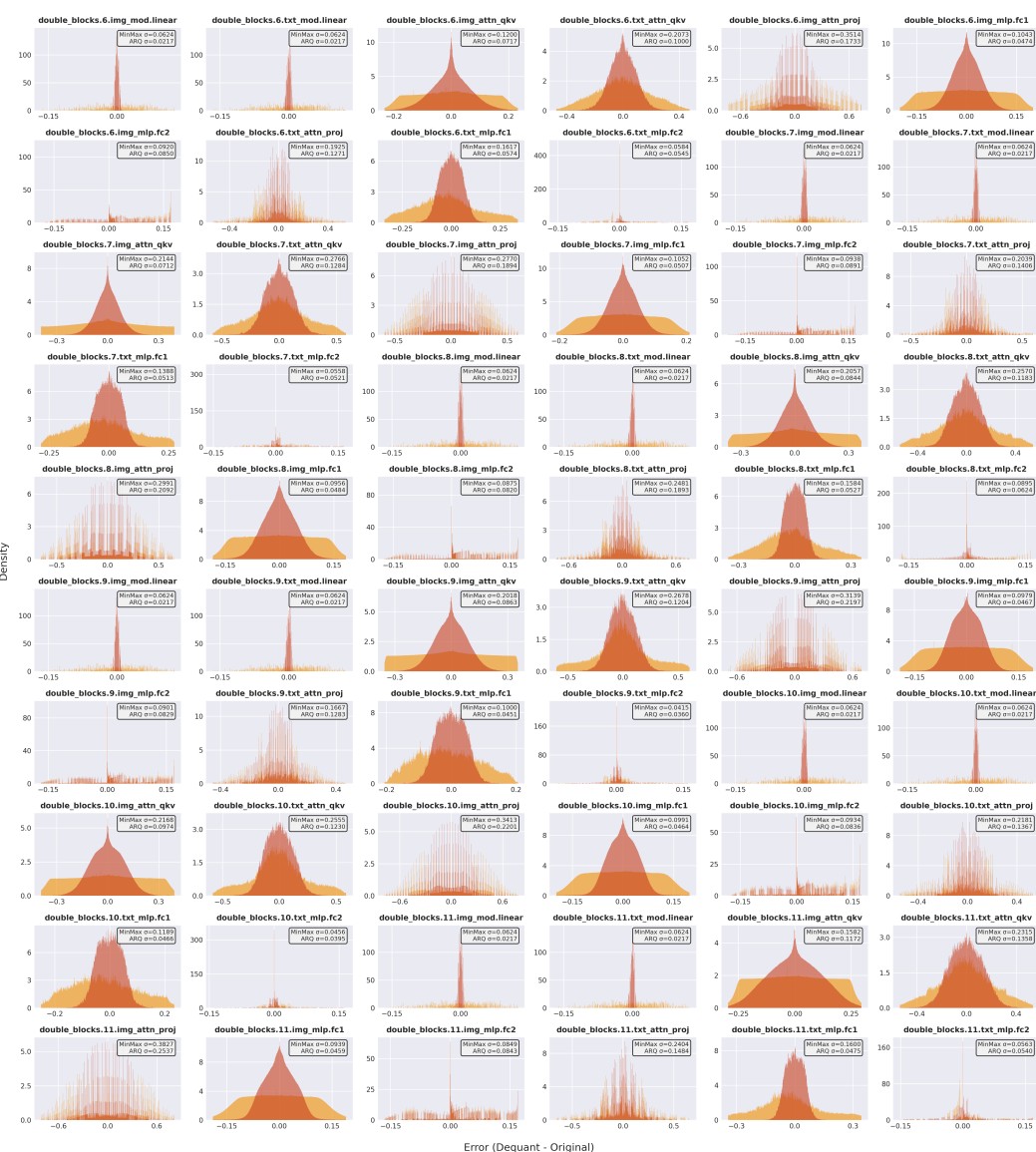

Figure 4: Quantization Error Distribution Comparison between MinMax (Jacob et al., 2018) and ARQ for several double blocks .

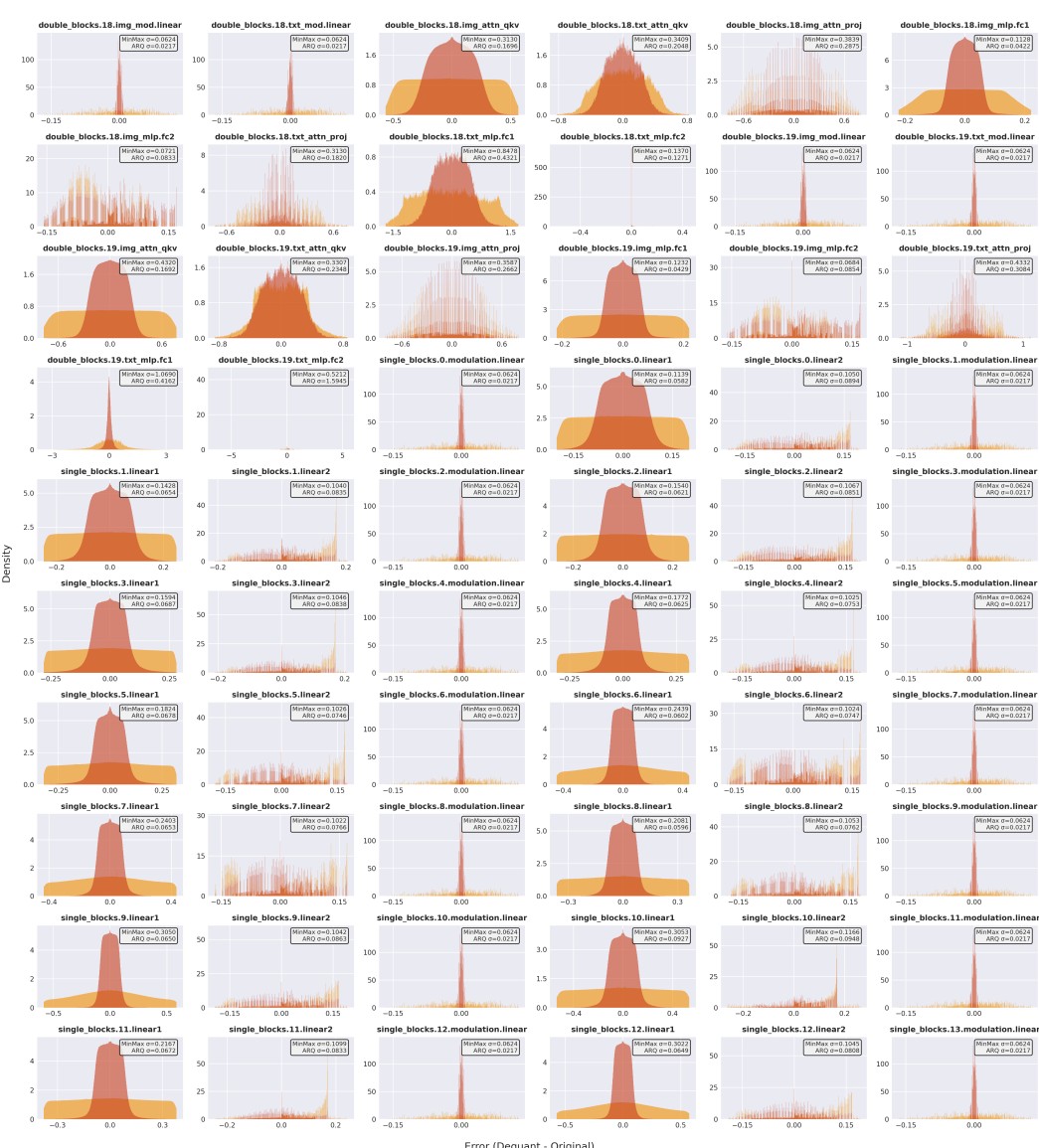

Figure 5: Quantization Error Distribution Comparison between MinMax (Jacob et al., 2018) and ARQ for several single blocks .

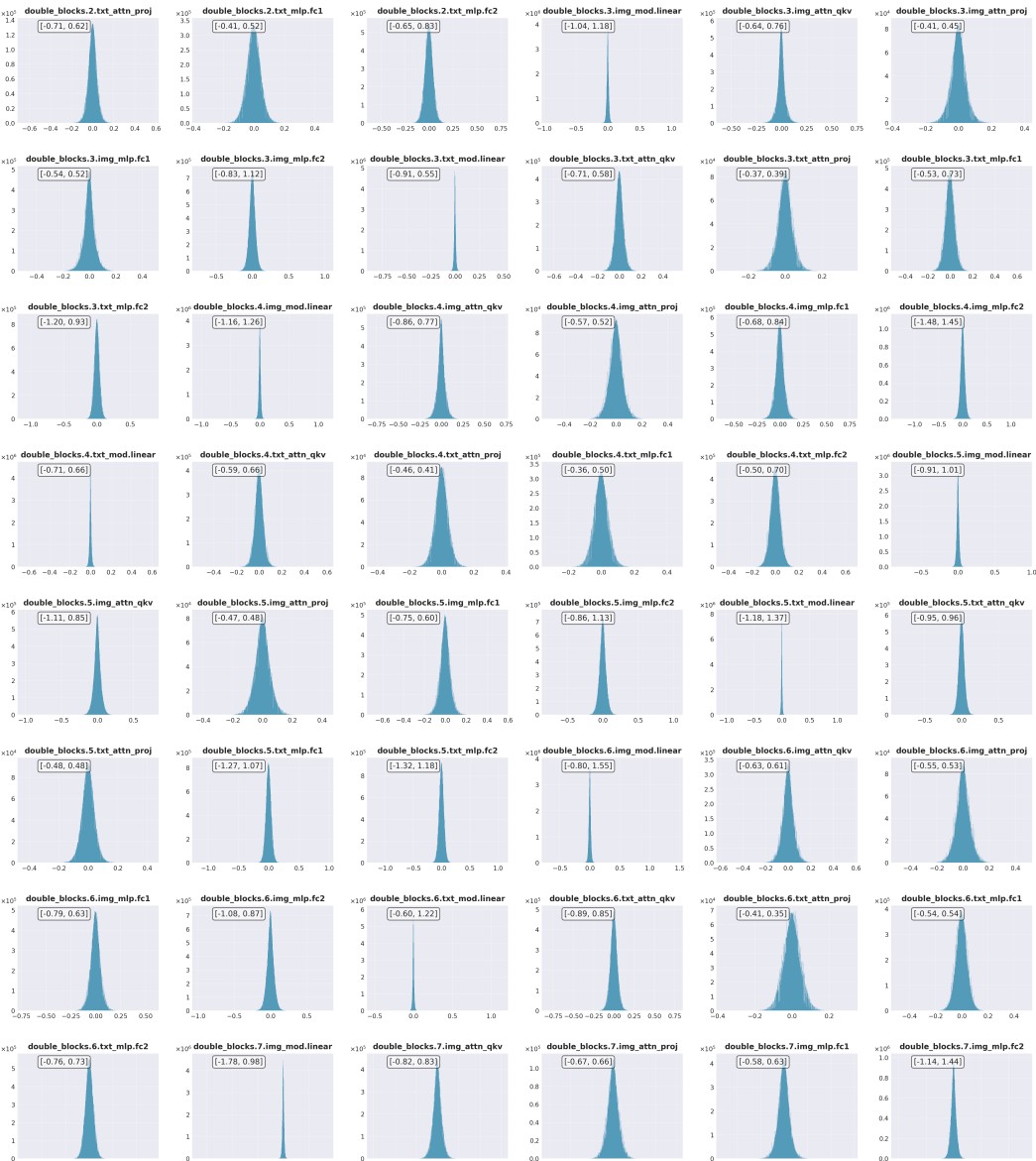

Figure 6: Weight Distribution of HunyuanVideo (Kong et al., 2024).

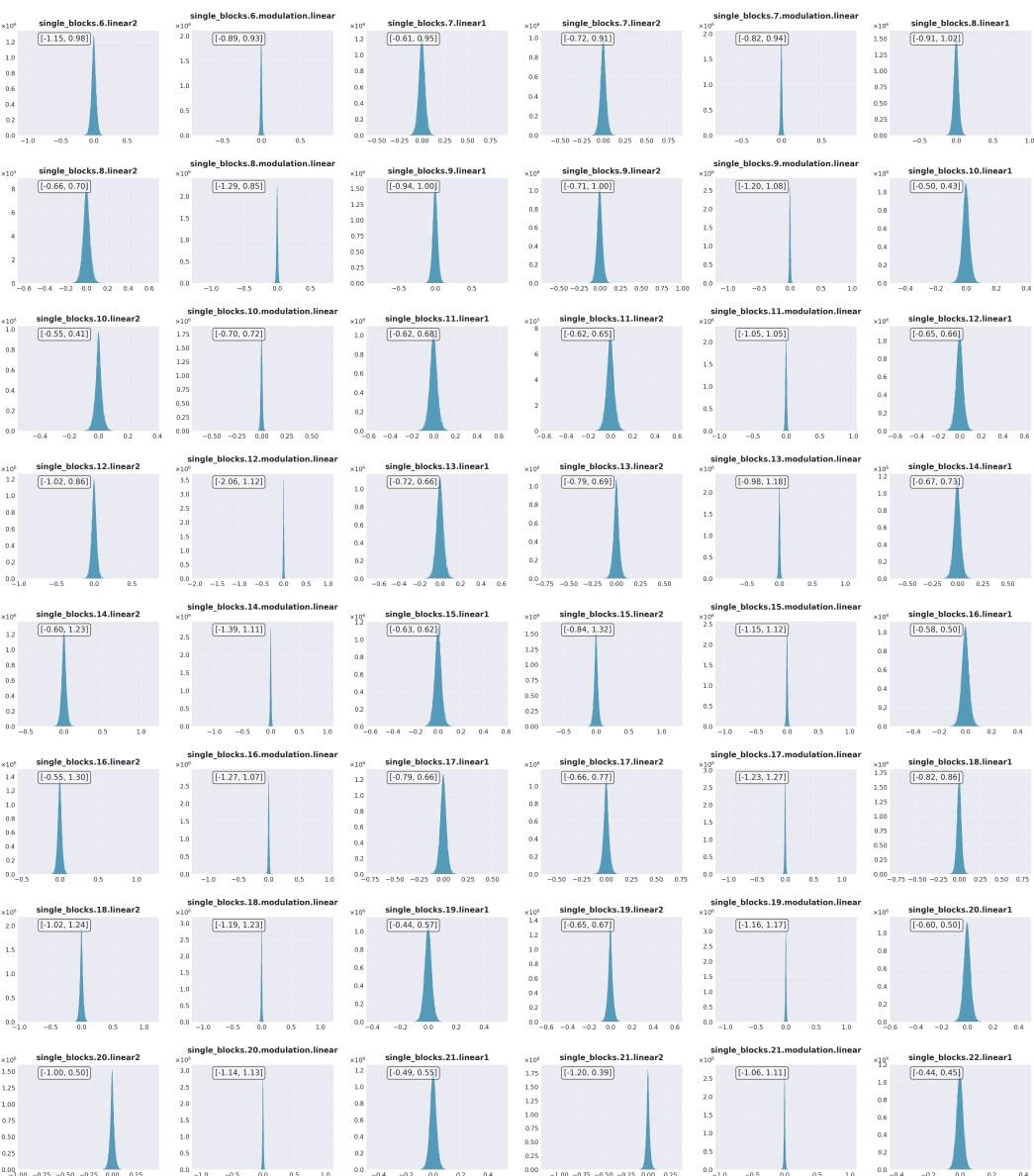

Figure 7: Weight Distribution of HunyuanVideo (Kong et al., 2024).