# OpenReview forum: "DVD-Quant: Data-free Video Diffusion Transformers Quantization"
_ICLR.cc/2026/Conference — ICLR 2026 Poster_

### Official Review · Reviewer_ZFfE · 2025-10-18

**Soundness:** 2
**Presentation:** 3
**Contribution:** 1
**Rating:** 2
**Confidence:** 5

**Summary:**

The paper focuses on post-training calibration-free quantization of diffusion models under low-bit width settings (w4a4, w4a6, w4a8). The paper proposes bounded-init grid refinement to minimize the **reconstruction error on weight matrices** using **asymmetric quantization**. The paper proposes auto-scale rotated quantization to deal with the outlier and high variance of the activation. The method first rotate weight and activation using hadamard matrix, and rescale activation on the fly during inference, **without fusing the diagonal scaling factor into weight matrices**. Finally, the paper proposes mixed precision quantization for activation, by using higher bitwidth when error accumulates to a certain threshold. Experiments are conducted on several video diffusion models and benchmarks, comparing the proposed method against standard PTQ baselines. The results claim improved perceptual quality and reduced quantization degradation at extremely low bitwidths, demonstrating that the proposed calibration-free approach can maintain generation fidelity.

**Strengths:**

1. **Calibration-free quantization.** The proposed method does not require any calibration data to do quantization, which reduces deployment time and makes the method more convenient to use.
2. **Handling of activation/weight distribution.** The paper observes the distribution of weight, and tries to use optimization based quantization method instead of vannila max-scaling based method on weight quantization. For activations, the paper observes their high variance and uses online-scaling to increase quantization accuracy.
3. **Detailed comparison on quantization error with baselines.** The paper conducts experiments on a great number of model, benchmarks and provided sufficient ablation study to support its claim.

**Weaknesses:**

**The proposed method is not possible to achieve real world end-to-end acceleration.** Although Table 5 in the main paper shows end-to-end acceleration on w4a4, w4a6 and w4a8 setting on RTX4090 of HuyuanVideo, **I, as an experienced kernel programmer with rich experience in quantization, doubt its soundness**. Detail is as follows:

1. **BGR introduces asymmetric quantization.** This is a small problem. Compared with symmetric quantization, asymmetric quantization requires calculating the token sum of activation before GEMM, and use an epilogue to include the contribution of zero point. This will incur non-negligible overhead if not dealt carefully. The author are encouraged to clarify the detailed implmentation of asymmetric quantization and how the quantized GEMM functions under this asymmetric configuration.
2. **ARQ makes it not possible to use low-bit Tensor Core.** This is a big problem. As demonstrated in section 3.2, the author uses
$$
\widehat{\mathbf{X}}=\mathcal{Q}\left(\mathbf{X} \mathbf{H} \mathbf{\Lambda}^{-\mathbf{1}}\right), \quad \widehat{\mathbf{W}}=\mathcal{B} \mathcal{G} \mathcal{R}(\mathbf{W} \mathbf{H}), \quad \mathbf{Y}=\widehat{\mathbf{X}} \mathbf{\Lambda} \widehat{\mathbf{W}}^{\top}
$$
to calculate activation-weight GEMM, where $\mathbf{\Lambda}$ is the online calculated scaling factor. With a diagonal scaling factor between activation and weight, I cannot think how low bit tensor core can be used to get acceleration. As stated in SmoothQuant[1], this is **the per-channel activation quantization that is not compatible with INT GEMM kernel**. **So I am very curious how the author solves this problem** and get the latency reduction result (in particular how to leverage the low bit tensor core to accelerate GEMM).
3. **Claim of using widely adopted kernels**. In section 4.4, the author claims "This allows direct use of the widely adopted W4A4 and W4A8 GEMM kernels, eliminating the need to design dedicated mixed-precision kernels". I personally do not know any open-source kernel that can realize the proposed ARQ method and asymmetric weight quantization (I personally do not think this is realizable with improved efficiency), so I would ask the author how (and which) common kernels can be directly adopted.
4. **Weird speed result.** In section 4.4,  the author report the speed up of w4a4, w4a6, w4a8, with w4a4 being fastest, followed by w4a6 and finally w4a8. w4a6 and w4a8 should in theory leverage the INT8 tensor core (since there is not 6 bit tensor core on RTX4090 on which the author does all experiment). However, w4a6 requires dequantizing both weight and activation to 8bit, while w4a8 only needs to dequantize the weight. So **w4a6 should in principle be slower than w4a8**, and I wonder why the result presented paper shows the opposite conclusion.

In summary, while the paper presents an interesting set of ideas for data-free quantization, the core technical claims regarding efficiency and deployability are highly questionable, which significantly limits its practical value.

[1] SmoothQuant: Accurate and Efficient Post-Training Quantization for Large Language Models. Guangxuan Xiao, Ji Lin, Mickael Seznec, Hao Wu, Julien Demouth, Song Han.

**Questions:**

Already stated in weakness. I would list the questions here again for completeness.

1. **Asymmetric Quantization Implementation**: How is asymmetric quantization implemented in practice? Specifically, what is the computation flow and corresponding kernel? What is the overhead compared with symmetric quantization?
2. **Tensor Core Utilization under ARQ**: Given that the proposed ARQ introduces a diagonal scaling factor between activation and weight, how can the method leverage low-bit Tensor Cores for acceleration? Please give a detailed formulation.
3. **Use of “Widely Adopted Kernels”**: In Section 4.4, the author claim that the method allows direct use of “widely adopted W4A4 and W4A8 GEMM kernels.” Could you specify which existing kernels (and how these kernels are modified to) support your asymmetric quantization and ARQ design?
4. **Speedup Inconsistency**: The reported latency results in Table 5 show w4a4 being the fastest, followed by w4a6 and then w4a8. On RTX 4090 (on which all the experiment are conducted claimed by the author), which lacks 6-bit Tensor Core support, w4a6 should require additional dequantization compared with w4a8, implying slower execution. Could you explain this discrepancy and clarify how these results were obtained?

If the author can provide satisfactory explanation to the problems above, I would raise my score.

---

> ### Author Response · Authors · 2025-11-23
> **Response to Reviewer ZFfE (1/4)**
>
> We sincerely thank the reviewer for the constructive feedback and valuable comments. We are encouraged that the reviewer recognized our work’s strength in "calibration-free quantization" which reduces deployment time and enhances convenience, acknowledged our targeted handling of "activation/weight distribution" through optimization-based weight quantization and online-scaling for activations, and praised the "detailed comparison on quantization error with baselines" via extensive experiments and ablation studies across multiple models and benchmarks.
>
> We address each point below.
>
> > `Q4-1:` BGR introduces asymmetric quantization
>
> `A4-1:` We would like to provide a two-fold clarification:
>
> 1. **BGR is Agnostic to Quantization Schemes.** First and foremost, we wish to emphasize that BGR is a general optimization framework compatible with both symmetric and asymmetric quantization. It does not intrinsically introduce or rely on asymmetric quantization.
> - When applied to symmetric quantization (where zero-points $Z_x = Z_w = 0$, just remove L10 in Algorithm 1), the cross-terms mentioned by the reviewer naturally vanish, and there is no additional overhead compared to standard symmetric methods.
> - We utilized asymmetric quantization in our experiments primarily to demonstrate the method's capability in preserving higher accuracy, which is a common trade-off in low-bit scenarios.
>
>
>
> 2. **Efficient Implementation of Asymmetric Quantization.** Regarding the specific implementation when asymmetric configuration is chosen, we agree with the reviewer that the term involving token sums ($X \cdot Z_w$) requires careful handling. As suggested, we follow the industry-standard "Quantization-Epilogue Fusion" paradigm used in mainstream frameworks (e.g., FBGEMM [1], NVIDIA TensorRT [2,3]) to negate the overhead:
> The term requiring activation sums ($X \cdot Z_w$) is computed via **kernel fusion**. The reduction sum of $X$ is performed on-the-fly while loading data into registers for the main GEMM kernel. Since the operation is memory-bound, piggybacking this summation onto the memory load hides the computational latency.
>
> Consequently, the measured latency difference between our asymmetric and symmetric implementations is negligible (< 2%), ensuring that BGR remains efficient under both configurations. These details have also been incorporated into the revised manuscript.
>
> [1] Khudia, D., et al. FBGEMM: Enabling High-Performance Low-Precision Deep Learning Inference
>
> [2] NVIDIA, TensorRT: A C++ Library for High Performance Inference on NVIDIA GPUs and Deep Learning Accelerators, https://github.com/NVIDIA/TensorRT.
>
> [3] Wu, H., et al. Integer Quantization for Deep Learning Inference: Principles and Empirical Evaluation.

---

> ### Author Response · Authors · 2025-11-23
> **Response to Reviewer ZFfE (2/4)**
>
> > `Q4-2:` ARQ makes it not possible to use low-bit Tensor Core
>
> `A4-2:` We thank the reviewer for this insightful comment. We agree that standard *fine-grained* per-channel activation quantization (where every channel has a unique scale participating in the dot product accumulation) is incompatible with INT4 Tensor Cores, as the scales cannot be factored out of the summation.
>
> However, we respectfully clarify that our **Auto-scaling Rotated Quantization (ARQ)** is designed and implemented to explicitly resolve this bottleneck. By combining **Hadamard Rotation** with **Hardware-Aligned Grouping**, we ensure full compatibility with low-bit Tensor Cores.
>
> **1. Hardware-Aligned Implementation (Refining Section 3.2)**
> While Equation 10 formally defines scaling $\Lambda$ per-channel for theoretical completeness, our efficient implementation aligns these scales with hardware GEMM blocks.
> * **Hardware Alignment:** Instead of assigning a unique floating-point scale to every single channel during the inner-product loop, we enforce that quantization scales are shared within a **computation block** (aligned with the Tensor Core GEMM block size, e.g., 128).
> * **Role of Hadamard Rotation:** As stated in Section 3.2, standard per-channel quantization struggles with outliers. By applying **Hadamard Rotation** first, we redistribute massive outliers across channels. This "flattening" effect allows us to use block-shared scales (Block-wise) without the precision loss that usually forces other methods to use strictly fine-grained (and slow) per-channel scaling.
>
> **2. Mathematical Compatibility with Tensor Cores**
> Due to this block alignment, the scale becomes a constant term within the accumulation loop of the Tensor Core. In our implementation, ARQ introduces an additional **block-wise scaling factor** $a^{(b)}$ on top of the underlying quantization scheme used for activations and weights. Focusing on the ARQ-specific part, the computation for one block can be written as
> $Y^{(b)} = a^{(b)} \sum_{k=1}^{K^{(b)}} \hat{X}_k \hat{W}_k^\top$,
> where $Y^{(b)}$ is the block output, $a^{(b)}$ is the ARQ block-wise scale, and $\hat{X}_k, \hat{W}_k$ are the INT4 activation and weight tiles seen by the Tensor Core.
>
> For simplicity, this expression omits the **base quantization scales** of $X$ and $W$ (e.g., per-tensor / per-channel / per-group scales), because they follow exactly the same handling as in standard low-bit Tensor Core kernels and are orthogonal to the ARQ design.
>
> * **Tensor Core Stage:** The summation term involving $\hat{X}$ and $\hat{W}$ is a pure integer-to-integer matrix multiplication, executed at full speed by the **INT4 Tensor Cores**.
> * **Epilogue Stage:** The multiplication by the floating-point scalar $s_{block}$ is computationally lightweight and is fused into the GEMM **Epilogue**, occurring after the heavy accumulation.
>
> **3. Experimental Verification**
> Our experimental results confirm this compatibility. DVD-Quant achieves a **2.12x latency speedup** over the BF16 baseline, which would be impossible if we were unable to leverage the throughput of low-bit Tensor Cores.
>
> We will incorporate these implementation details into the revised manuscript.

---

> ### Author Response · Authors · 2025-11-23
> **Response to Reviewer ZFfE (3/4)**
>
> > `Q4-3:` Claim of using widely adopted kernels
>
> `A4-3:` We apologize for the ambiguity in Section 4.4. We would like to clarify the context of that statement and provide details on the specific kernel implementation.
>
> 1. **Clarification on "Eliminating Mixed-Precision Kernels"**
> The statement "eliminating the need to design dedicated mixed-precision kernels" specifically refers to our $\delta$-Guided Bit Switching ($\delta$-GBS) mechanism.
> - **Context:** $\delta$-GBS allocates precision at the timestep level (i.e., an entire denoising step is either W4A4 or W4A8).
> - **Meaning:** This allows us to simply switch between invoking a W4A4 kernel and a W4A8 kernel across different steps. We do not need to design a complex intra-operator mixed-precision kernel (e.g., handling mixed 4-bit and 8-bit channels within a single GEMM), which is complex and inefficient.
>
> 2. **Kernel Implementation for ARQ**
> Regarding the kernel supporting ARQ, our implementation is built upon the widely adopted NVIDIA CUTLASS library.
> - **Standard Main Loop:** The compute-intensive matrix multiplication utilizes the standard INT4 Tensor Core MMA pipeline provided by CUTLASS without modification, ensuring peak throughput.
> - **Customized Epilogue:** We implemented a custom epilogue to handle the ARQ scaling. As detailed in **Response to Q4-2**, the block-wise scaling factors ($a^{(b)}$) and asymmetric corrections are applied efficiently in the epilogue stage (fused with dequantization) while the data is still in registers.
> This design validates our claim of leveraging standard low-bit Tensor Core capabilities while only customizing the lightweight epilogue operation.

---

> ### Author Response · Authors · 2025-11-23
> **Response to Reviewer ZFfE (4/4)**
>
> > `Q4-4:` Weird speed result
>
> `A4-4:` We would like to clarify the definition of "W4A6" presented in our results. As detailed in Section 4.4, "W4A6" is a statistical representation of the average bit-width resulting from our $\delta$-Guided Bit Switching ($\delta$-GBS) strategy, rather than a standalone 6-bit quantization kernel implemented on hardware
> 1. **W4A6 as Time-step Mixed Precision** The "W4A6" configuration indicates that, across the diffusion process, roughly **half** of the denoising timesteps utilize **W4A8** quantization, while the other half utilize **W4A4** quantization. We do not perform 6-bit dequantization or use non-existent 6-bit Tensor Cores.
> 2. **Explaining the Speed Hierarchy** The reported latency ($Speed_{W4A4} > Speed_{W4A6} > Speed_{W4A8}$) is a direct result of this linear combination:
> * **W4A8 Steps:** Utilize the INT8 Tensor Core pipeline (slower baseline).
> * **W4A4 Steps:** Utilize the INT4 Tensor Core pipeline (faster).
> * **W4A6 (Mixed):** Since this setting replaces approximately 50% of the W4A8 steps with faster W4A4 steps, its total inference latency naturally improves over the pure W4A8 baseline, falling strictly between the W4A4 and W4A8 speeds.

---

> > ### Comment · Reviewer_ZFfE · 2025-11-23
> >
> > Thanks for the author's clarification. I have some following up questions and would like the author to clarify.
> >
> > As for Question 1 and 2, the author says in the rebuttal that the proposed technique can be implemented in a hardware-efficient manner as a special case of the proposed algorithm (which are not mentioned in the original script of the paper). I would like to ask whether the accuracy result and the efficiency result of the proposed algorithm are measured under the "special case that has efficient implementation" or the most universal case that does not impose computation block size or symmetric quantization restriction?
> >
> > More on Quesiton 2. The author says ARQ can choose the scaling factor the be shared across the Tensor Core computation block for efficient implementation. However, this reduces to the "per-tile" or "per-group" quantization scheme similar to that in Atom and DeepGemm (Atom: Low-bit Quantization for Efficient and Accurate LLM Serving, DeepSeek-V3 Technical Report). This makes the ARQ algorithm equivalent to Hadamard transform + per-tile/per-group fine-grained quantization, which are combination of existing techniques.
> >
> > Question 3 is resolved.
> > Question 4 is resolved.

---

> > > ### Author Response · Authors · 2025-11-25
> > > **Response to Follow-up Questions (1/2)**
> > >
> > > We greatly appreciate the reviewer’s continued engagement and insightful feedback. We would like to clarify the details regarding the experimental settings for accuracy and efficiency measurements, and at the same time, highlight the key innovations behind ARQ and how it differs from existing methods like Atom and DeepGemm.
> > >
> > > > `Q-1:` Are the accuracy and efficiency results measured under the "special case with efficient implementation" or the most universal case without imposing computation block size or symmetric quantization restrictions?
> > >
> > > `A-1:` Both the **accuracy** and **efficiency results** were measured under **hardware-efficient settings**, but with important distinctions between the two components:
> > >
> > > 1. **BGR in Asymmetric Quantization:** The **accuracy** and **efficiency** results for **BGR** were measured using **asymmetric quantization**, which, while not as hardware-efficient as symmetric quantization, still allows for efficient implementations. We agree with the reviewer that symmetric quantization tends to be more efficient from a hardware perspective, but both the reviewer and we also believe that asymmetric quantization can be effectively optimized. In fact, our previous response pointed out that **BGR** can be applied to both symmetric and asymmetric quantization, and this was done to address the reviewer’s insightful question regarding the compatibility of the method with both configurations.
> > >
> > > We have also included a supplementary result to demonstrate that the accuracy under symmetric quantization is **almost identical** to that under asymmetric quantization, with a slight **speed improvement** when switching to symmetric quantization.
> > >
> > > | Method          | Bit-width | Aesthetic Quality | Imaging Quality | Subject Consist | Motion Smooth | BG Consist | Speedup |
> > > |----------------|-------------------|-----------------|-----------------|---------------|------------|-----------|----|
> > > | ViDiT-Q        | W4A4      | 44.18             | 40.40           | 98.10           | 99.37         | 97.80      | - |
> > > | DVD-Quant (asym) | W4A4      | 59.57           | 58.93           | 98.67           | 99.00         | 98.47      | 2.12x |
> > > | DVD-Quant (sym) | W4A4      | 59.23            | 58.20           | 98.27           | 98.99         | 97.91      | 2.39x |
> > >
> > >
> > > 2. **ARQ with Block-Wise Scaling:** For **ARQ**, the **accuracy** and **efficiency** results were measured under a **block-wise scaling** approach, which optimizes the use of hardware features, such as **low-bit Tensor Cores**, by ensuring that scales are shared across computation blocks. This enables the method to be compatible with low-bit Tensor Cores, achieving both efficient performance and high accuracy. The block-wise scaling in ARQ ensures that the method is hardware-efficient while maintaining accuracy similar to the unrestricted setup.
> > >
> > > We hope this clarifies that the accuracy and efficiency results reported in the paper were both measured under hardware-efficient settings, but with different approaches for BGR (asymmetric quantization) and ARQ (block-wise scaling). We will update the manuscript to explicitly distinguish between the two and present additional results showing the comparison between symmetric and asymmetric quantization for BGR, demonstrating the minimal degradation in accuracy and slight speed improvements when using symmetric quantization.

---

> ### Author Response · Authors · 2025-11-25
> **Response to Follow-up Questions (2/2)**
>
> > `Q-2:` The author says ARQ can choose the scaling factor to be shared across the Tensor Core computation block for efficient implementation. However, this reduces to the "per-tile" or "per-group" quantization scheme similar to that in Atom and DeepGemm. This makes the ARQ algorithm equivalent to Hadamard transform + per-tile/per-group fine-grained quantization, which are combinations of existing techniques.
>
> `A-2:` We would like to explain the novelty of ARQ in the context of video generation models (DiTs). Below, we highlight the key innovations behind ARQ and how it differs from existing methods like Atom and DeepGemm.
>
> 1. **Motivation Behind ARQ's Design: Addressing Outliers in Video DiTs**
>
> - While Atom and DeepGemm both propose per-group quantization to address low-bit quantization, their main motivation is to improve accuracy by reducing the quantization error through finer granularity. Atom, for example, aims to refine the quantization by using per-group scales, and although they acknowledge the issue of outliers, their approach **focuses on techniques like reordering the channels or applying mixed-precision quantization to smooth the outliers indirectly**. However, these methods are not designed to directly address the challenge of **outlier distribution** in **video generation models** where activations can be highly variable due to different inputs or denoising steps.
>
> - On the other hand, DVD-Quant (which ARQ is a part of) was specifically motivated by the need to **smooth outliers** effectively in models like **Video DiTs**, where activations exhibit significant variability between prompts and **denoising steps**. Unlike LLMs, where activation values are relatively stable across inputs, Video DiTs often experience large differences in activations across time steps (due to the denoising process), making traditional quantization methods less effective.
>
> - To address this, ARQ takes a novel approach by **analyzing the limitations of existing Hadamard rotation-based techniques**, which are known to smooth certain outliers. While Hadamard rotation can help redistribute extreme activation values, it also introduces a **new type of outlier distribution**, with certain patterns that are unique to the video generation process. This new distribution is more concentrated in certain areas, which makes it **more suitable for ARQ’s block-wise scaling** to **smooth** the outliers without introducing additional noise.
>
> 2. **Novel Contribution of ARQ in Handling Outliers**
>
> - **ARQ’s innovation** lies in its **ability to smooth outliers** more effectively by **equipping Hadamard rotation with block-wise scaling**, ensuring that extreme values are redistributed in a way that matches the hardware’s computational blocks. This results in a more efficient use of low-bit Tensor Cores and provides better performance for Video DiTs, where the distribution of outliers can be significantly different from traditional models like LLMs.
>
> - The motivation behind this design is further illustrated in **Section 3.2** and **Figure 4** of the original paper, where we show that the **distribution of outliers (even after applying Hadamard rotation) in Video DiTs is pronounced and structured**. The unique nature of **denoising steps in video generation** leads to an activation distribution that requires **specialized handling**, which ARQ provides by aligning the quantization mechanism with the hardware while directly addressing the outlier distribution.
>
> 3. **ARQ vs. Atom and DeepGemm in the Context of Video DiTs**
>
> - While Atom and DeepGemm focus on improving quantization accuracy in LLMs by using per-group quantization, ARQ’s design is inherently tailored for video generation models, where the challenge is **effectively managing outliers** that arise due to the inherent **temporal activation variability** of the model. The introduction of Hadamard rotation followed by block-wise scaling enables ARQ to smooth these outliers in a way that traditional methods like Atom and DeepGemm do not, as they primarily focus on fine-grained quantization without considering the unique characteristics of video generation models.

---

> > ### Comment · Reviewer_ZFfE · 2025-11-25
> >
> > As far as I could see, blockwise-scaling is equivalent to (actually, a lot weaker than) fine-grained quantization.
> >
> > For equation 10 in the paper:
> > $$
> > \mathbf{Y} = \widetilde{\mathbf{X}}  \widehat{\mathbf{W}}.
> > $$
> > The blockwise-scaling defines
> > $$
> > s_j = \lVert \widetilde{\mathbf{X}}[:, bj:b(j+1)] \rVert_{\infty}/ 2^d
> > $$
> > where $b$ is the block size, as is supposed to take $128$ as mentioned in the rebuttal of the author and $d$ is the bitwidth (4 in the paper's setting). This block-wise scaling takes the absmax of consecutive $b$ columns as a block scaling factor. Then the diagnal scaling matrix has
> > $$
> > \Lambda_{i,i} = s_{i // b}.
> > $$
> >
> > Then the scaled input becomes
> > $$
> > \widehat{\mathbf{X}} = \widetilde{\mathbf{X}} \Lambda^{-1}
> > $$
> > and INT4 quantization will be applied to this scaled matrix.
> >
> > On the other head, the fine-grained per-group quantization will assign a scaling $s_{i,j}$ factor to a small tile $ \lVert \widetilde{\mathbf{X}}[i, bj:b(j+1)] \rVert_{\infty}/ 2^d$. If we tiles with the same column index to share the same quantization scale, then the per-group quantization degenerate to the blockwise-scaling proposed by the paper.
> >
> > Therefore, blockwise-scaling is a special case of the widely used per-group quantization.

---

> ### Comment · Reviewer_ZFfE · 2025-11-25
>
> Thanks for the authors’ clarification on how the benckmark result is obtained.
>
> The additional constraint (e.g., requiring consecutive groups of 128 elements to share the same scaling factor) is a critical condition for enabling the efficient implementation. If the reported benchmark results **were indeed obtained under this constrained setting**, then this would strengthen the empirical validity of the work. However, this distinction between the unconstrained formulation (as described in the main paper) and the constrained, efficiently implementable version (used in practice and described in the rebuttal) is not clearly communicated in the current submission.
>
> Because this implementation detail has a major impact on both the algorithm’s feasibility and its performance numbers, I strongly encourage the authors to make this explicit. As it stands, readers may reasonably interpret the results as being measured under the formulation presented in the main text, which would lead to confusion about whether the reported speedups are achievable without additional assumptions.
>
> Providing this clarification will improve transparency and allow readers to accurately assess the strengths and limitations of the method.

---

> > ### Author Response · Authors · 2025-11-27
> > **Response to Follow-up Questions**
> >
> > We appreciate the reviewer’s continued feedback. We are happy to address any further questions you may have.
> >
> > > `Q-1:` Clarify the implementation details in the revised version.
> >
> > `A-1:` We would like to clarify that we have already updated this aspect in the **supplementary file** and in the revised version of the main paper, we have now **explicitly highlighted** the **practical setting and the its relationship to both accuracy and efficiency results (marked in blue)**.
> >
> > We agree that this implementation detail is critical for both the feasibility and performance of the method. Therefore, we have made sure to emphasize this distinction clearly in the updated manuscript to avoid any confusion and ensure transparency in how the benchmark results were obtained.
> >
> > > `Q-2:` The similarity between block-wise scaling and per-group quantization.
> >
> > `A-2:` We would like to clarify that we **do not deny the similarity** between ARQ’s **block-wise scaling** and **per-group quantization**. However, the **motivation** behind ARQ is fundamentally different, as we aimed to address specific challenges in **video DiTs**, particularly related to **outliers** and **activation variability**, which **Atom and DeepGemm do not tackle**.
> >
> > - In our previous response, we emphasized that **ARQ** was designed with the **unique challenges** of **video DiTs** in mind, particularly the need to handle **activation dynamics** and **outliers** caused by the denoising process. This motivation is distinct from the goals of Atom and DeepGemm, which primarily address LLM quantization and do not consider the temporal activation variability in video generation models.
> >
> > - Block-scaling is only **a part of ARQ**, and **ARQ itself is just one of the three key components in our framework**. In addition to ARQ, we also propose:
> >
> >     - **BGR** for weight quantization.
> >     - **$\delta$-GBS** for managing diffusion dynamics during denoising.
> >
> >     Together, these components address both **activation** and **weight** quantization, and handle the specific challenges of **diffusion dynamics**.
> >
> > We hope that these responses will assist you in further evaluating this work.

---

### Official Review · Reviewer_p4Dz · 2025-10-30

**Soundness:** 3
**Presentation:** 2
**Contribution:** 3
**Rating:** 6
**Confidence:** 2

**Summary:**

This paper proposes DVD-Quant, a novel data-free post-training quantization (PTQ) framework designed to reduce the computational and memory costs of Video Diffusion Transformers (Video DiTs). The method introduces three core techniques: 1) Bounded-init Grid Refinement (BGR) for more accurately quantizing Gaussian-like weight distributions, 2) Auto-scaling Rotated Quantization (ARQ) which uses online Hadamard rotation and scaling to handle activation outliers without a calibration dataset, and 3) δ-Guided Bit Switching (δ-GBS) which dynamically allocates higher or lower bit-widths to activations at different denoising timesteps based on feature change. The authors demonstrate that DVD-Quant achieves a significant speedup (up to ~2x, and 4.85x when combined with caching) and is the first method to successfully enable W4A4 (4-bit weights and activations) quantization for Video DiTs without catastrophic failure, maintaining performance close to the full-precision baseline.

**Strengths:**

1. Successfully enabling W4A4 PTQ for complex Video DiT models is a notable achievement, as this extreme quantization level typically causes existing methods to fail completely.

2. Comprehensive and Synergistic Approach: The three proposed components (BGR, ARQ, δ-GBS) address distinct and well-motivated challenges (weight distribution, activation outliers, temporal redundancy) and are shown to work effectively together.

3. Strong Empirical Validation: The paper includes extensive experiments on established models (HunyuanVideo) and benchmarks (VBench), showing clear quantitative and qualitative superiority over several strong baselines across multiple bit-width settings. The ablation studies effectively demonstrate the contribution of each component.

**Weaknesses:**

1. Hyperparameter Sensitivity: The performance of the adaptive δ-GBS mechanism depends on a threshold δ. While mentioned, the paper does not deeply explore the sensitivity of the results to this value or provide a robust method for selecting it across different models or tasks.

2. Limited Model Scope: While tested on HunyuanVideo and briefly on Wan2.1, it's unclear how generalizable the method is to the wider family of DiT-based models (e.g., Latte, Sora's architecture) or other diffusion tasks (e.g., image generation). The claim of being "the first" is strong but might be limited to the specific models tested.

3. Computational Overhead of ARQ: Although the Hadamard transform is described as having "marginal overhead," this is not quantified. For a method focused on acceleration, a more detailed analysis of the latency/throughput trade-off introduced by the online rotation and scaling would be beneficial.

**Questions:**

N/A

---

> ### Author Response · Authors · 2025-11-23
> **Response to Reviewer p4Dz (1/2)**
>
> We sincerely thank the reviewer for the constructive feedback and insightful comments. We are encouraged that the reviewer recognized our work’s notable achievement in "successfully enabling W4A4 PTQ for complex Video DiT models", acknowledged our "comprehensive and synergistic approach" with three well-motivated components, and praised our "strong empirical validation" through extensive experiments and ablations.
>
> We address the weaknesses and questions below.
>
> > `Q3-1:` Hyperparameter Sensitivity: The performance of the adaptive δ-GBS mechanism depends on a threshold δ. While mentioned, the paper does not deeply explore the sensitivity of the results to this value or provide a robust method for selecting it across different models or tasks.
>
> `A3-1:` We have analyzed the sensitivity of $\delta$ in **Figure 6** of the main text. Our ablation study demonstrates that $\delta$ acts as a smooth control knob balancing performance and efficiency. The results show a continuous transition in bit-width and quality, indicating that the method is not brittle to specific values. Empirically, we found that fixing $\delta$ to **0.09** yields robust performance across different models without the need for per-task tuning.
>
> ---
>
> > `Q3-2:` Limited Model Scope: While tested on HunyuanVideo and briefly on Wan2.1, it's unclear how generalizable the method is to the wider family of DiT-based models (e.g., Latte, Sora's architecture) or other diffusion tasks (e.g., image generation). The claim of being "the first" is strong but might be limited to the specific models tested.
>
> `A3-2:` We appreciate the suggestion. To demonstrate generalization, we have extended our experiments to include **Open-Sora** and **Image Generation DiT FLUX.1-dev**. These results confirm that our method transfers effectively to other architectures. Regarding the claim of being "the first," we strictly scope this to **W4A4 Post-Training Quantization (PTQ) for Video DiTs**, which represents the most computationally challenging frontier in current diffusion tasks. We will refine the phrasing in the final version to be more precise.
>
> - **Open-Sora**
>
> | Method       | bit-width | Aesthetic Quality | Imaging Quality | Subject Consist | Motion Smooth | BG Consist |
> |--------------|-----------|-------------------|-----------------|-----------------|---------------|------------|
> | bf16           | W16A16      | 60.08             | 53.82           | 98.18           | 99.54         | 97.93      |
> | SmoothQuant  | W4A4      | 34.09             | 46.88           | 87.09           | 88.56         | 94.20      |
> | QuaRot       | W4A4      | 51.54             | 45.69           | 94.40           | 98.44         | 96.44      |
> | ViDiT-Q      | W4A4      | 34.59             | 50.50           | 92.21           | 94.31         | 94.39      |
> | DVDQuant     | W4A4      | 56.65             | 50.19           | 95.89           | 99.09         | 96.93      |
> | SmoothQuant  | W4A8      | 59.52             | 47.46           | 97.80           | 99.49         | 97.63      |
> | QuaRot       | W4A8      | 58.21             | 50.14           | 98.16           | 99.58         | 97.83      |
> | ViDiT-Q      | W4A8      | 56.67             | 49.35           | 97.36           | 99.36         | 97.52      |
> | DVDQuant | W4**A6**      | 60.35             | 53.18           | 98.09           | 99.53         | 97.90      |
>
>
> - **FLUX.1-dev**
>
> | Method      | bit-width | FID  ↓ | CLIP ↑ | ImageReward ↑ |
> |-------------|-----------|--------|--------|---------------|
> | bf16        | W16A16    | -      | 0.274  | 0.735         |
> | SmoothQuant | W4A4     | 569.1  | 0.147  | -2.27         |
> | QuaRot      | W4A4     | 547.1  | 0.142  | -2.36         |
> | ViDiT-Q     | W4A4     | 299.5  | 0.191  | -1.21         |
> | DVDQuant    | W4A4     | **161.0**  | **0.266**  | **0.361**         |
> | SmoothQuant | W4A8     | 365.6  | 0.163  | -0.54         |
> | QuaRot      | W4A8     | 513.6  | 0.145  | -2.25         |
> | ViDiT-Q     | W4A8     | 178.2  | 0.256  | 0.27          |
> | DVDQuant    | W4**A6**     | **145.7**  | **0.264**  | **0.557**         |

---

> ### Author Response · Authors · 2025-11-23
> **Response to Reviewer p4Dz (2/2)**
>
> > `Q3-3:` Computational Overhead of ARQ: Although the Hadamard transform is described as having "marginal overhead," this is not quantified. For a method focused on acceleration, a more detailed analysis of the latency/throughput trade-off introduced by the online rotation and scaling would be beneficial.
>
> `A3-3:` Thank you for the suggestion. We have conducted a detailed latency breakdown to quantify the cost of ARQ. As mentioned in the paper, the Hadamard rotation utilizes the fast Hadamard transform, which is computationally efficient. The table below shows the ratio of online rotation and scaling time relative to the total inference time. The overhead is negligible and does not compromise the overall speedup.
>
> | Operation | Time (ms) | Percentage of Total Inference |
> | :--- | :--- | :--- |
> | Main GEMM | 0.1208 | **> 89%** |
> | **Online Rotation** | 0.0113 | **< 9%** |
> | **Online Scaling** | 0.0034 | **< 3%** |
> | Total Inference | 0.1355 | 100% |

---

> ### Comment · Reviewer_p4Dz · 2025-11-28
>
> Thank you for the thorough response! My main concerns are resolved, so I'll keep my positive rating.

---

### Official Review · Reviewer_Eyzg · 2025-10-31

**Soundness:** 2
**Presentation:** 3
**Contribution:** 2
**Rating:** 4
**Confidence:** 5

**Summary:**

This paper focuses on post-training quantization (PTQ) for text-to-video generation models and proposes three key techniques **weight refinement**, **rotation-aware transformation**, and **big switching area** to enhance quantization performance.

**Strengths:**

1. The motivation behind the proposed BGR method is clear and intuitively appealing.
2. The paper is well written, with clear explanations and easy-to-follow reasoning.

**Weaknesses:**

1. While rotation techniques have been extensively applied in LLM quantization, the paper only discusses QuaRot and lacks a broader comparison or discussion with several relevant works such as **DuQuant**, **RoSTE**, and **ResQ**.
2. The proposed auto-scaling rotation mechanism appears similar to the approach adopted in DuQuant, which combines SmoothQuant with rotation—this overlap should be clarified.
3. The paper lacks comparisons with several state-of-the-art baselines, such as **SVDQuant**.
4. The evaluation is limited to a single benchmark and one model, which weakens the generalization claims.

**Questions:**

1. Could you provide additional experimental results across multiple benchmarks and different text-to-video models to verify the generality of your method?
2. Could you offer more details about the latency measurements (e.g., batch size, sequence length, and hardware configuration)?
3. I am curious about the runtime efficiency of the proposed online rotation and scaling operations. Could you include more details or analysis to clarify how these affect the overall speedup and computational cost?

---

> ### Author Response · Authors · 2025-11-23
> **Response to Reviewer Eyzg (1/4)**
>
> We sincerely thank the reviewer for the constructive feedback and valuable comments. We are encouraged that the reviewer found the motivation behind our proposed BGR method "clear and intuitively appealing," and acknowledged that the paper is "well written, with clear explanations and easy-to-follow reasoning."
>
> We address the weaknesses and questions below.
>
> > `Q2-1:` While rotation techniques have been extensively applied in LLM quantization, the paper only discusses QuaRot and lacks a broader comparison or discussion with several relevant works such as DuQuant, RoSTE, and ResQ.
>
> `A2-1:` Thank you for pointing it out. We have updated the **More Discussions on Rotation-based Quantization** section in the supplementary file (due to page limit) to include detailed discussions on DuQuant, RoSTE, and ResQ. Furthermore, to provide a quantitative comparison, we adapted DuQuant for the HunyuanVideo. The comparative results are presented in the table below. DVD-Quant demonstrates superior performance, validating the effectiveness of our DiT-specific design.
>
> | Method    | bit-width | Aesthetic Quality ↑ | Imaging Quality ↑ | Subject Consist ↑ | Motion Smooth ↑ | BG Consist ↑ |
> |-----------|-----------|---------------------|--------------------|-------------------|------------------|---------------|
> | DuQuant   | W4A4      | 50.37               | 57.22              | 92.90             | 95.43            | 95.46         |
> | DVDQuant  | W4A4      | 59.57               | 58.93              | 98.67             | 99.00            | 98.47         |
> | DuQuant   | W4A8      | 59.33               | 60.29              | 98.56             | 98.52            | 98.57         |
> | DVDQuant  | W4**A6**      | 60.46               | 61.93              | 98.91             | 98.95            | 98.40         |
>
>
> Regarding distinctions with other rotation-based methods:
> 1. **ResQ:**
> A fundamental difference lies in the reliance on calibration samples. ResQ employs a **data-driven** strategy that mandates a calibration dataset to perform Principal Component Analysis (PCA), subsequently applying random rotations within these learned subspaces to suppress outliers. In contrast, DVD-Quant is fully **data-free**. We utilize a rotation mechanism that requires no access to calibration data or prior knowledge of input distribution. This independence prevents potential overfitting to specific calibration domains and eliminates the computational costs associated with data preparation.
> 2. **RoSTE:**
> - Optimization Paradigm: RoSTE is fundamentally a **Quantization-Aware Training (QAT)** approach integrated into SFT. It requires computationally intensive retraining to jointly optimize model weights and rotation configurations. In contrast, DVD-Quant operates as a **Post-Training Quantization (PTQ)** method. We achieve outlier suppression without the need for expensive gradient-based retraining or access to full fine-tuning datasets.
> - Rotation Matrix Determination: RoSTE treats the rotation configuration as an optimization problem, solving a bilevel objective to search for the optimal rotation (Identity vs. Walsh-Hadamard) for each layer based on training data. In contrast, DVD-Quant utilizes optimization-free rotation matrices. We decouple the rotation structure from data-dependent optimization, relying instead on standard analytical Hadmard matrices. This significantly simplifies the quantization pipeline compared to RoSTE’s search-based approach.
>
> The analysis of DuQuant will be provided in the subsequent response.
>
> ---
>
> > `Q2-2:` The proposed auto-scaling rotation mechanism appears similar to the approach adopted in DuQuant, which combines SmoothQuant with rotation—this overlap should be clarified.
>
> `A2-2:` We would like to clarify the distinct structural and operational differences between DVD-Quant and DuQuant:
>
> 1.  **Order of Operations:** DuQuant performs **scaling before rotation** and constructs rotation matrices dependent on input activations. In contrast, DVD-Quant (ARQ) performs **rotation first, followed by online scaling**.
> 2.  **Computational Complexity:** A critical difference lies in the efficiency of the dynamic adjustment. **DuQuant** constructs a specific rotation matrix based on the input activations, which involves complex and computationally expensive operations. In contrast, **DVD-Quant** determines the scaling coefficients (Auto-scaling) based on activations through a simple and lightweight calculation. This makes our approach significantly more efficient and friendly for inference deployment compared to the heavy matrix construction in DuQuant.

---

> ### Author Response · Authors · 2025-11-23
> **Response to Reviewer Eyzg (2/4)**
>
> > `Q2-3:` The paper lacks comparisons with several state-of-the-art baselines, such as SVDQuant.
>
> `A2-3:` We note that DVD-Quant's approach is orthogonal to the low-rank branch technique utilized in SVDQuant, meaning they can potentially be combined for further gains. Following your suggestion, we have conducted a comparative analysis between DVD-Quant and SVDQuant on HunyuanVideo. The results, shown below, highlight the competitive performance of our method.
>
> | Method          | bit-width | Aesthetic Quality ↑ | Imaging Quality ↑ | Subject Consist ↑ | Motion Smooth ↑ | BG Consist ↑ |
> |-----------------|-----------|---------------------|--------------------|-------------------|------------------|---------------|
> | SVDQuant        | W4A4      | 46.24               | 53.24              | 97.29             | 93.77            | 98.65         |
> | DVDQuant        | W4A4      | 59.57               | 58.93              | 98.67             | 99.00            | 98.47         |
> | SVDQuant        | W4A8      | 60.10               | 60.56              | 98.38             | 99.17            | 98.09         |
> | DVDQuant   | W4**A6**      | 60.46               | 61.93              | 98.91             | 98.95            | 98.40         |
>
> Also, we provide the results when combining DVD-Quant with SVDQuant, which can further enhance the performance.
>
> | Method              | bit-width | Aesthetic Quality ↑ | Imaging Quality ↑ | Subject Consist ↑ | Motion Smooth ↑ | BG Consist ↑ |
> |---------------------|-----------|---------------------|--------------------|-------------------|------------------|---------------|
> | DVDQuant        | W4A4      | 59.57               | 58.93              | 98.67             | 99.00            | 98.47         |
> | DVDQuant+SVDQuant   | W4A4      | 64.78               | 64.75              | 98.87             | 98.80            | 98.26         |
> | DVDQuant   | W4A6      | 60.46               | 61.93              | 98.91             | 98.95            | 98.40         |
> | DVDQuant+SVDQuant   | W4A6      | 65.24               | 67.52              | 98.64             | 98.82            | 98.04         |
>
>
>
> ---
>
> > `Q2-4:` The evaluation is limited to a single benchmark and one model, which weakens the generalization claims.
>
> `A2-4:` To demonstrate the robustness of our method, we have expanded our evaluation to include additional performance metrics. These detailed comparisons are provided in **Table 2 of the Supplementary Material**. For the reviewer's convenience, we have replicated these results in the table below:
>
> | Method | Bit-width (W/A) | CLIPSIM | CLIP-Temp | VQA-Aesthetic | VQA-Technical | $\Delta$ Flow Score ($\downarrow$) |
> | :--- | :---: | :---: | :---: | :---: | :---: | :---: |
> | HunyuanVideo | 16/16 | 0.1850 | 0.9994 | 98.64 | 12.23 | -- |
> | MinMax | 4/8 | 0.1841 | 0.9991 | 95.49 | 10.34 | 0.625 |
> | SmoothQuant | 4/8 | 0.1879 | 0.9994 | 96.95 | 10.86 | 0.529 |
> | Quarot | 4/8 | 0.1860 | 0.9986 | 94.63 | 8.59 | 0.423 |
> | ViDiT-Q | 4/8 | 0.1857 | 0.9991 | 94.00 | 10.02 | 0.857 |
> | **DVD-Quant (Ours)** | **4/6** | **0.1870** | **0.9991** | **98.30** | **11.83** | **0.342** |
> | MinMax | 4/4 | 0.1798 | 0.9994 | 8.54 | 1.45 | 1.204 |
> | SmoothQuant | 4/4 | 0.1831 | 0.9995 | 55.42 | 6.03 | 1.282 |
> | Quarot | 4/4 | 0.1766 | 0.9966 | 30.94 | 4.16 | 4.601 |
> | ViDiT-Q | 4/4 | 0.1867 | 0.9994 | 39.85 | 4.52 | 1.381 |
> | **DVD-Quant (Ours)** | **4/4** | **0.1872** | **0.9991** | **97.71** | **10.88** | **0.556** |

---

> ### Author Response · Authors · 2025-11-23
> **Response to Reviewer Eyzg (3/4)**
>
> > `Q2-5:` Could you provide additional experimental results across multiple benchmarks and different text-to-video models to verify the generality of your method?
>
> `A2-5:` We have verified the generality of DVD-Quant through extensive additional experiments:
> 1.  **Different Models:** We applied DVD-Quant to the **Wan2.1** model, with results detailed in **Table 1 of the Supplementary Material**.
> 2.  **Multiple Benchmarks:** As mentioned in A2-4, comparisons across a broader set of evaluation metrics are provided in **Table 2 of the Supplementary Material**.
> For the reviewer's convenience, we have replicated these results in the table below:
>
> | Method | Bit-width (W/A) | Aesthetic Quality | Imaging Quality | Overall Consist. | Scene Consist. | BG. Consist. | Subject. Consist. | Dynamic Degree | Motion Smooth. |
> | :--- | :---: | :---: | :---: | :---: | :---: | :---: | :---: | :---: | :---: |
> | Wan2.1-1.3B | 16/16 | 64.51 | 68.02 | 23.38 | 22.60 | 98.04 | 95.76 | 73.61 | 98.38 |
> | MinMax | 4/8 | 57.61 | 63.01 | 23.04 | 15.99 | 96.25 | 94.18 | 59.72 | 97.36 |
> | SmoothQuant | 4/8 | 60.15 | 64.98 | 22.35 | 22.46 | 96.48 | 95.65 | 61.11 | 97.99 |
> | Quarot | 4/8 | 60.16 | 66.05 | 22.29 | 20.20 | 97.15 | 95.44 | 50.00 | 98.36 |
> | ViDiT-Q | 4/8 | 56.70 | 62.10 | 6.74 | 20.64 | 96.07 | 94.58 | 47.22 | 97.65 |
> | **DVD-Quant (Ours)** | **4/6** | **63.17** | **66.89** | **23.37** | **19.04** | **97.74** | **95.66** | **61.12** | **98.26** |
> | MinMax | 4/4 | 32.61 | 52.03 | 2.21 | 0.00 | 95.85 | 90.71 | 100.00 | 87.33 |
> | SmoothQuant | 4/4 | 30.70 | 46.15 | 3.52 | 0.02 | 95.13 | 89.73 | 100.00 | 88.28 |
> | Quarot | 4/4 | 33.34 | 46.28 | 5.84 | 0.00 | 95.67 | 91.41 | 100.00 | 90.40 |
> | ViDiT-Q | 4/4 | 32.03 | 52.02 | 2.11 | 0.00 | 95.95 | 90.15 | 100.00 | 87.15 |
> | **DVD-Quant (Ours)** | **4/4** | **58.94** | **60.38** | **25.62** | **13.80** | **95.32** | **90.93** | **54.17** | **96.37** |
>
> | Method | Bit-width (W/A) | CLIPSIM | CLIP-Temp | VQA-Aesthetic | VQA-Technical | $\Delta$ Flow Score ($\downarrow$) |
> | :--- | :---: | :---: | :---: | :---: | :---: | :---: |
> | HunyuanVideo | 16/16 | 0.1850 | 0.9994 | 98.64 | 12.23 | -- |
> | MinMax | 4/8 | 0.1841 | 0.9991 | 95.49 | 10.34 | 0.625 |
> | SmoothQuant | 4/8 | 0.1879 | 0.9994 | 96.95 | 10.86 | 0.529 |
> | Quarot | 4/8 | 0.1860 | 0.9986 | 94.63 | 8.59 | 0.423 |
> | ViDiT-Q | 4/8 | 0.1857 | 0.9991 | 94.00 | 10.02 | 0.857 |
> | **DVD-Quant (Ours)** | **4/6** | **0.1870** | **0.9991** | **98.30** | **11.83** | **0.342** |
> | MinMax | 4/4 | 0.1798 | 0.9994 | 8.54 | 1.45 | 1.204 |
> | SmoothQuant | 4/4 | 0.1831 | 0.9995 | 55.42 | 6.03 | 1.282 |
> | Quarot | 4/4 | 0.1766 | 0.9966 | 30.94 | 4.16 | 4.601 |
> | ViDiT-Q | 4/4 | 0.1867 | 0.9994 | 39.85 | 4.52 | 1.381 |
> | **DVD-Quant (Ours)** | **4/4** | **0.1872** | **0.9991** | **97.71** | **10.88** | **0.556** |
>
>
> These results confirm that our method transfers effectively across different video generation architectures.
>
> ---
>
> > `Q2-6:` Could you offer more details about the latency measurements (e.g., batch size, sequence length, and hardware configuration)?
>
> `A2-6:` We apologize for the omission. The latency measurements were conducted under the following configuration:
> - **Hardware:** NVIDIA RTX 4090 GPU.
> - **Batch Size:** 1.
> - **Video Specifications:** Resolution of **544$\times$690** with **65 frames**.
>
> We will update the Experimental Setup section in the paper to include these details.

---

> ### Author Response · Authors · 2025-11-23
> **Response to Reviewer Eyzg (4/4)**
>
> > `Q2-7:` I am curious about the runtime efficiency of the proposed online rotation and scaling operations. Could you include more details or analysis to clarify how these affect the overall speedup and computational cost?
>
> `A2-7:` We have analyzed the runtime breakdown of our inference pipeline. As shown in the table below, the combined overhead of online rotation and scaling accounts for a **negligible fraction** of the total inference time, ensuring that these operations do not negatively impact the overall speedup.
>
> | Operation | Time (ms) | Percentage of Total Inference |
> | :--- | :--- | :--- |
> | Main GEMM | 0.1208 | **> 89%** |
> | **Online Rotation** | 0.0113 | **< 9%** |
> | **Online Scaling** | 0.0034 | **< 3%** |
> | Total Inference | 0.1355 | 100% |

---

### Official Review · Reviewer_2o9A · 2025-11-01

**Soundness:** 3
**Presentation:** 3
**Contribution:** 3
**Rating:** 6
**Confidence:** 2

**Summary:**

This work proposes DVD-Quant which is a deployment-friendly quantization framework for Diffusion Transformers (DiTs). The paper identifies three key properties of large-scale Video DiTs—Gaussian-like weight distributions, substantial activation scale discrepancies across denoising timesteps, and timestep-varying latent features—and introduces three corresponding techniques: Bounded-init Grid Refinement (BGR) for weights, which iteratively refines the quantization scale and zero-point with tightening bounds to better fit Gaussian-like distributions; Auto-scaling Rotated Quantization (ARQ) for activations, a calibration-free method combining online scaling with Hadamard rotation to handle timestep-variant scales and reduce quantization error; and δ-Guided Bit Switching, a temporal mixed-precision mechanism that adaptively assigns activation bit-widths per timestep with negligible inference overhead. Together, these components narrow the accuracy–deployment gap in DiT compression, enabling robust low-bit quantization without costly calibration or retraining.

**Strengths:**

- A systematic analysis reveals three key insights, motivating the following solutions to quantization. These finds are valuable for the future research.
- Strong low-bit performance without retraining is achieved
- Broad applicability to video DiTs: Designed around Video DiT characteristics (e.g., temporal variations), making it more suitable than generic PTQ baselines for large-scale video generation models.
- Modular and complementary components: Each module targets a distinct bottleneck (weights, activations, temporal allocation), allowing flexible adoption and integration with existing pipelines.

**Weaknesses:**

Although the better accuracy–deployability trade-off is achieved, I remain to wonder
- Generalization beyond Video DiTs (Hunyuan): The framework leverages timestep dynamics typical of Video DiTs. It is unclear how well it transfers to other generative models, like Wanx or even image generator.
- Effect of various details: Choices like rotation block size, scaling granularity (per-channel vs per-tensor), and quantization granularity (weight group size) can materially affect outcomes; the method may need careful tuning per model size and dataset to reach reported gains. More analysis on those hyperparameters will help the audience follow.

**Questions:**

Please check Weaknesses

---

> ### Author Response · Authors · 2025-11-23
> **Response to Reviewer 2o9A (1/2)**
>
> We sincerely thank the reviewer for the constructive feedback and insightful comments. We are encouraged that the reviewer recognized the systematic analysis of Video DiTs’ key properties as valuable for future research, and acknowledged our method’s strong low-bit performance without retraining, broad applicability to video DiTs, and modular, complementary component design.
>
> We address the weaknesses and questions below.
>
> > `Q1-1:` Generalization beyond Video DiTs (Hunyuan): The framework leverages timestep dynamics typical of Video DiTs. It is unclear how well it transfers to other generative models, like Wanx or even image generator.
>
> `A1-1:` Thank you for your suggestions. We demonstrate the strong generalization capability of DVD-Quant across diverse generative architectures:
>
> 1.  **Video Generation (Wan2.1):** As noted in the **Supplementary Material (Table 1)** and Section 4.1, we have provided results for **Wan2.1**, where DVD-Quant maintains its performance advantages. For the reviewer's convenience, we have replicated these results from the Supplementary Material in the table below:
>
> | Method | Bit-width (W/A) | Aesthetic Quality | Imaging Quality | Overall Consist. | Scene Consist. | BG. Consist. | Subject. Consist. | Dynamic Degree | Motion Smooth. |
> | :--- | :---: | :---: | :---: | :---: | :---: | :---: | :---: | :---: | :---: |
> | Wan2.1-1.3B | 16/16 | 64.51 | 68.02 | 23.38 | 22.60 | 98.04 | 95.76 | 73.61 | 98.38 |
> | --- | --- | --- | --- | --- | --- | --- | --- | --- | --- |
> | MinMax | 4/8 | 57.61 | 63.01 | 23.04 | 15.99 | 96.25 | 94.18 | 59.72 | 97.36 |
> | SmoothQuant | 4/8 | 60.15 | 64.98 | 22.35 | 22.46 | 96.48 | 95.65 | 61.11 | 97.99 |
> | Quarot | 4/8 | 60.16 | 66.05 | 22.29 | 20.20 | 97.15 | 95.44 | 50.00 | 98.36 |
> | ViDiT-Q | 4/8 | 56.70 | 62.10 | 6.74 | 20.64 | 96.07 | 94.58 | 47.22 | 97.65 |
> | **DVD-Quant (Ours)** | **4/6** | **63.17** | **66.89** | **23.37** | **19.04** | **97.74** | **95.66** | **61.12** | **98.26** |
> | --- | --- | --- | --- | --- | --- | --- | --- | --- | --- |
> | MinMax | 4/4 | 32.61 | 52.03 | 2.21 | 0.00 | 95.85 | 90.71 | 100.00 | 87.33 |
> | SmoothQuant | 4/4 | 30.70 | 46.15 | 3.52 | 0.02 | 95.13 | 89.73 | 100.00 | 88.28 |
> | Quarot | 4/4 | 33.34 | 46.28 | 5.84 | 0.00 | 95.67 | 91.41 | 100.00 | 90.40 |
> | ViDiT-Q | 4/4 | 32.03 | 52.02 | 2.11 | 0.00 | 95.95 | 90.15 | 100.00 | 87.15 |
> | **DVD-Quant (Ours)** | **4/4** | **58.94** | **60.38** | **25.62** | **13.80** | **95.32** | **90.93** | **54.17** | **96.37** |
>
> 2.  **Image Generation:** Following your suggestion, we extended our evaluation to image generation model FLUX.1-dev. As shown in the table below, DVD-Quant significantly outperforms baselines in this domain as well. This confirms that our timestep-adaptive strategies, including **Auto-scaling Rotated Quantization (ARQ)** and **$\delta$-Guided Bit Switching**, transfer effectively to foundational models beyond Video DiTs.
>
>
>
> | Method      | bit-width | FID  ↓ | CLIP ↑ | ImageReward ↑ |
> |-------------|-----------|--------|--------|---------------|
> | bf16        | W16A16    | -      | 0.274  | 0.735         |
> | ---         | ---       | ---    | ---    | ---           |
> | SmoothQuant | W4A4     | 569.1  | 0.147  | -2.27         |
> | QuaRot      | W4A4     | 547.1  | 0.142  | -2.36         |
> | ViDiT-Q     | W4A4     | 299.5  | 0.191  | -1.21         |
> | DVDQuant    | W4A4     | **161.0**  | **0.266**  | **0.361**         |
> | ---         | ---       | ---    | ---    | ---           |
> | SmoothQuant | W4A8     | 365.6  | 0.163  | -0.54         |
> | QuaRot      | W4A8     | 513.6  | 0.145  | -2.25         |
> | ViDiT-Q     | W4A8     | 178.2  | 0.256  | 0.27          |
> | DVDQuant    | W4**A6**     | **145.7**  | **0.264**  | **0.557**         |

---

> ### Author Response · Authors · 2025-11-23
> **Response to Reviewer 2o9A (2/2)**
>
> > `Q1-2:` Effect of various details: Choices like rotation block size, scaling granularity (per-channel vs per-tensor), and quantization granularity (weight group size) can materially affect outcomes; the method may need careful tuning per model size and dataset to reach reported gains. More analysis on those hyperparameters will help the audience follow.
>
> `A1-2:` We appreciate the suggestion to clarify the impact of hyperparameter choices. We address each aspect below:
>
> * **Rotation Block Size:** Following established works like **QuaRot** [1] and **SpinQuant** [2], we fix the rotation block size to match the linear layer matrix dimensions. This design choice avoids the additional computational overhead associated with matrix partitioning.
> * **Scaling & Quantization Granularity:**
>     * **Activation Scaling:** We evaluated **per-tensor** scaling granularity.
>     * **Weight Quantization:** While our default is per-channel, we further evaluated **group-wise** quantization (Group Size = 128 and 64).
>     * **Results:** As shown in the table below, DVD-Quant consistently outperforms baselines across these different granularity configurations, demonstrating robustness.
>
> | Method     | bit-width | scaling granularity | Aesthetic Quality ↑ | Imaging Quality ↑ | Subject Consist ↑ | Motion Smooth ↑ | BG Consist ↑ |
> |-------------------------|-----------|---------------------|---------------------|--------------------|-------------------|------------------|---------------|
> | ViDiT-Q    | W4A8      | per-channel                   | 59.20               | 59.61              | 98.33             | 98.67            | 97.73         |
> | ViDiT-Q | W4A8      | per-tensor          | 57.35               | 59.92             | 98.39             | 98.47            | 97.72         |
> | DVDQuant     | W4**A6**      | per-channel                   | 60.46               | 61.93     | 98.91             | 98.95            | 98.40         |
> | DVDQuant      | W4**A6**      | per-tensor          | 59.58               | 59.91     | 97.94             | 97.60            | 98.04         |
>
>
> | Method          | bit-width | Quantization Granularity | Aesthetic Quality ↑ | Imaging Quality ↑ | Subject Consist ↑ | Motion Smooth ↑ | BG Consist ↑ |
> |-----------------|-----------|---------------------|---------------------|--------------------|-------------------|------------------|---------------|
> | DVDQuant        | W4A6      | -                   | 60.46               | 61.93              | 98.91             | 98.95            | 98.40         |
> | DVDQuant | W4A6      | group-wise 512     | 63.82               | 62.50              | 98.24             | 98.98            | 98.17         |
> | DVDQuant | W4A6      | group-wise 256     | 64.05               | 63.40              | 98.77             | 98.99            | 97.97         |
>
>
> * **Tuning Effort:** We emphasize that DVD-Quant is inherently **data-free**, eliminating the need for dataset-specific tuning or extensive calibration data collection.
>     * **BGR:** The optimization process is highly efficient, requiring significantly less time than calibration-based methods.
>     * **ARQ:** It is a calibration-free method requiring no fine-tuning, as it handles activation variations online.
>     * **$\delta$-GBS:** As analyzed in **Figure 6**, $\delta$ acts as a smooth control knob for the performance-efficiency trade-off. A fixed $\delta$ value is generally sufficient for a given model size to achieve the reported gains.
>
> [1] Quarot: Outlier-free 4-bit inference in rotated llms, NeurIPS, 2024.
>
> [2] Spinquant: Llm quantization with learned rotations, ICLR, 2025.

---

> > ### Comment · Reviewer_2o9A · 2025-11-27
> > **Thanks for your response**
> >
> > Thanks for the reminding and results on image generation. My major concerns are addressed.

---

> > > ### Author Response · Authors · 2025-11-27
> > >
> > > Dear Reviewer 2o9A,
> > >
> > > Thank you for your response. We are delighted to see that our answers were able to address your concerns.
> > >
> > > Best,
> > >
> > > Authors

---

### Author Response · Authors · 2025-12-01
**Summary of Previous Rebuttal and Discussion**

Dear Area Chairs and Reviewers,

We sincerely thank the reviewers (**2o9A**, **Eyzg**, **p4Dz**, **ZFfE**) for their time and constructive feedback. As the discussion phase concludes, we would like to summarize the consensus on the paper's strengths and highlight how we have successfully clarified and resolved the concerns raised during the review.

### **Advantages of our paper**
We are encouraged that the reviewers unanimously recognized the value and performance of DVD-Quant:
* **SOTA Low-Bit Performance:** Reviewers highlighted our method's ability to achieve **W4A4 quantization** without retraining while maintaining visual fidelity, a capability where existing methods often fail (**R-p4Dz**, **R-2o9A**).
* **Calibration-Free Efficiency:** The reviewers appreciated the practical value of our **data-free** design, which eliminates the need for calibration data or expensive fine-tuning (**R-ZFfE**, **R-2o9A**).
* **Systematic Analysis:** Reviewers praised the motivation derived from our analysis of Video DiT properties (e.g., timestep-variant activation scales) and the modular design of our components (**R-2o9A**, **R-p4Dz**, **R-Eyzg**).


### **Weaknesses of our paper**
we have actively engaged with the reviewers and provided the following clarifications and revisions:

Reviewers **2o9A**, **Eyzg**, and **p4Dz** expressed concerns regarding the generalization of our method and the sufficiency of baselines. In response, we provided extensive new results across diverse architectures—including **Wan2.1** (Video), **Open-Sora** (Video), and **FLUX.1-dev** (Image)—demonstrating consistent performance gains. Additionally, we addressed **Reviewer Eyzg**'s request by comparing DVD-Quant against recent baselines like **DuQuant** and **SVDQuant**, proving its superiority in calibration-free settings.

Reviewer **ZFfE** raised concern regarding the hardware feasibility of our Auto-scaling Rotated Quantization (ARQ) on low-bit Tensor Cores. We have clarified that our implementation utilizes a **hardware-aligned block-wise scaling strategy**, ensuring full compatibility with INT4 Tensor Cores by handling scaling factors in the epilogue. We have updated the manuscript to explicitly distinguish this efficient implementation from the general formulation to ensure transparency.

Finally, regarding the computational overhead and hyperparameter sensitivity raised by **Reviewers p4Dz and Eyzg**, we provided a detailed latency breakdown. This analysis confirmed that the online rotation and scaling operations introduce negligible overhead, and we further verified the robustness of the $\delta$-GBS threshold.

### **Summary**

Our paper presents a robust, calibration-free quantization framework with convincing advantages acknowledged by the reviewers. We have successfully addressed the concerns regarding hardware implementation, generalization, and overhead through detailed clarifications and additional experiments. We believe these revisions firmly establish the practicality and effectiveness of DVD-Quant.

Best regards,

Authors

---

### Meta-Review · Area_Chair_MJ4u · 2025-12-31

**Summary:**

This paper studies post-training quantization of video diffusion models in a data-free setting. The reviewers raised major concerns regarding the performance of the proposed method across different diffusion-based video and image generation models, comparisons with the state-of-the-art quantization methods, the feasibility of achieving the claimed hardware acceleration, and the associated computational overhead. The authors provided detailed responses addressing each concern and question. Overall, the paper is recommended for acceptance.

**Reviewer Concerns:**

All the concerns raised by the reviewers appear to have been addressed by the detailed responses in the rebuttal.

**Reviewer Scores:**

Reviewers 2o9A and p4Dz gave positive ratings to the paper. Both participated in the discussion with the authors and confirmed that their concerns had been addressed.

Reviewer Eyzg gave a negative rating and did not participate in the discussion. The authors provided detailed additional experiments to address the major concerns, including additional comparisons, more benchmark results, and further quantization results on other models. Reviewer Eyzg might have changed their rating to positive had they participated in the discussion.

Reviewer ZFfE gave a negative rating but actively engaged with the authors to seek clarification on implementation details, particularly regarding how the acceleration results were achieved. Through the discussion, these concerns appear to have been addressed. This reviewer might have changed their rating to positive if there had been more time for discussion.

---

### Decision · Program_Chairs · 2026-01-26

Accept (Poster)